# Neutrophils and galectin-3 defend mice from lethal bacterial infection and humans from acute respiratory failure

Sudipta Das[1], Tomasz W. Kaminski[2,9], Brent T. Schlegel [3,9], William Bain [1,4,9], Sanmei Hu[1], Akruti Patel[1], Sagar L. Kale[1], Kong Chen[1], Janet S. Lee [5], Rama K. Mallampalli [6], Valerian E. Kagan [7], Dhivyaa Rajasundaram[3], Bryan J. McVerry [1], Prithu Sundd[2], Georgios D. Kitsios [1,10], Anuradha Ray [1,8,10] & Prabir Ray [1,8,10] ✉

Respiratory infection by *Pseudomonas aeruginosa*, common in hospitalized immunocompromised and immunocompetent ventilated patients, can be life-threatening because of antibiotic resistance. This raises the question of whether the host's immune system can be educated to combat this bacterium. Here we show that prior exposure to a single low dose of lipopolysaccharide (LPS) protects mice from a lethal infection by *P. aeruginosa*. LPS exposure trained the innate immune system by promoting expansion of neutrophil and interstitial macrophage populations distinguishable from other immune cells with enrichment of gene sets for phagocytosis- and cell-killing-associated genes. The cell-killing gene set in the neutrophil population uniquely expressed *Lgals3*, which encodes the multifunctional antibacterial protein, galectin-3. Intravital imaging for bacterial phagocytosis, assessment of bacterial killing and neutrophil-associated galectin-3 protein levels together with use of galectin-3-deficient mice collectively highlight neutrophils and galectin-3 as central players in LPS-mediated protection. Patients with acute respiratory failure revealed significantly higher galectin-3 levels in endotracheal aspirates (ETAs) of survivors compared to non-survivors, galectin-3 levels strongly correlating with a neutrophil signature in the ETAs and a prognostically favorable hypoinflammatory plasma biomarker subphenotype. Taken together, our study provides impetus for harnessing the potential of galectin-3-expressing neutrophils to protect from lethal infections and respiratory failure.

*Pseudomonas aeruginosa* is a Gram-negative bacterium that can be cleared by host defense mechanisms from the respiratory tract of healthy individuals but can cause pneumonia with a high degree of morbidity and mortality in immunocompromised patients[1–3]. *P. aeruginosa* is also a leading cause of acute infections resulting in ventilator-associated pneumonia (VAP) in hospitalized immunocompetent

patients[4] and can cause chronic infections in cystic fibrosis patients[5]. Infections by *P. aeruginosa* can be life-threatening and difficult to treat because the bacterium can acquire wide range antibiotic resistance through various genetic mechanisms[6]. Therefore, it is important to consider alternate avenues to fight this pathogen, such as by harnessing the host's immune response.

---

 1

Bacterial lipopolysaccharide (LPS), present in the outer membrane of Gram-negative bacteria, is an immunomodulatory agent[7]. An early study showed the ability of LPS to induce resistance to subsequent infections by different Gram-negative bacteria[8]. LPS-induced resistance results in reduced bacterial burden with promotion of leukocyte recruitment[9]. These findings suggest that LPS can equip the immune system to better defend the host against the invading pathogen. However, the mechanisms underlying the protective effects of LPS are not well understood. Here we show that prior exposure to a low dose of LPS protects mice from a lethal infection by *P. aeruginosa*. Pre-exposure to LPS in PA-infected mice causes accumulation of a neutrophil population uniquely enriched in both phagocytosis-associated genes and a cell-killing gene set comprising Lgals3, which encodes the antibacterial protein, galectin-3. Functional studies in mice support the impact of LPS on neutrophils in conferring protection. In humans, survivors of acute respiratory failure show significantly higher galectin-3 levels in endotracheal aspirates (ETAs) compared to non-survivors, galectin-3 levels strongly correlating with a neutrophil signature in the ETAs and a prognostically favorable hypoinflammatory plasma biomarker subphenotype.

## Results

### Pre-exposure to LPS protects mice from a lethal infection by *P. aeruginosa* with signature of a regulated inflammatory response

We sought to determine whether LPS can protect mice from a lethal infection by PA14, a strain commonly used to study *P. aeruginosa* infection of mice[10,11]. Mice intratracheally (i.t.) infected with $10^6$ colony forming units (cfu) of PA14 reproducibly appeared huddled and ill by 4−6 h after infection. We first assessed bacterial load in the lung tissue and in the peripheral blood by infecting mice on day 1 or 3 after i.t. administration of a low dose of 20 μg of ultrapure LPS. All mice were sacrificed at 4 h post-infection, a time point at which infected mice that were not pre-exposed to LPS appeared visibly ill (Fig. 1a). A significantly lower bacterial load was detected in the lungs of mice after exposure to LPS compared to that in mice that did not receive any LPS (Fig. 1b). No dissemination of bacteria in the blood was observed in LPS-treated mice whether infected on day 1 or 3 after LPS exposure, which was clearly evident in the control infected mice, all mice sacrificed at 4 h after infection (Fig. 1c). The mice infected with PA14 without LPS pre-exposure appeared near death after 18−20 h and were euthanized (Fig. 1d). This time point was considered the time of death for mice not pre-exposed to LPS since our Institutional Animal Care and Use Committee (IACUC) rules do not permit death as an end-point in survival studies. In contrast, all PA14-infected mice that received LPS 3 days earlier behaved normally without displaying any evidence of sickness and this was followed up to 14 days after infection (Fig. 1d). LPS-mediated protection was observed in both sexes. Thus, these experiments established $10^6$ cfu as a lethal dose of infection and the two groups of mice with or without LPS pre-exposure are hereafter referred to as LPS + PA14 and PA14 respectively. Bronchoalveolar lavage fluid (BALF) albumin concentration, indicative of alveolar permeability, was significantly higher in the PA14 mice compared to that in the LPS ± PA14 mice (Fig. 1e).

We measured mRNA levels of key genes associated with host defense by RT-qPCR and also assayed cytokine and chemokine levels along with markers of neutrophil accumulation, critical for defense against PA[12], in the lung. The RT-qPCR data revealed differential expression of infection-response genes between the LPS + PA14 and PA14 mice. The PA14 mice mounted a significantly greater expression of the cytokine genes (*Il1b*, *Il6*) compared to the LPS + PA14 mice (Fig. 1f). In contrast, the expression of the interferon stimulated genes (ISGs), *Ifit1* and *Isg15*, and *Ifng* were significantly higher in the LPS + PA14 mice compared to that in the PA14 mice. The expression of the anti-inflammatory cytokine, *Il10*, was attenuated in the LPS + PA14 mice, albeit expressed above baseline (Fig. 1f). We also examined the

expression of *Stat1*, a transcription factor activated by both Type I and II interferons (IFNs), which was higher in the LPS + PA14 mice compared to that in the PA14 mice (Fig. 1f). The profile of gene expression in the LPS-treated mice prompted us to also assess the expression of the enzyme Acyloxyacyl hydrolase (*Aoah*) that specifically facilitates the removal of secondary fatty acyl chains from the lipid A moiety of LPS thereby limiting the toxic effects of LPS and yet allowing an immune response after LPS exposure[13–15]. *Aoah* expression was low in both naïve and PA14-infected mouse lungs but its expression was significantly higher in response to LPS, with highest expression in the LPS + PA14 mice on day 3 post-LPS exposure (Fig. 1f) when the lung bacterial load was the lowest (Fig. 1b). We assayed mediators present in the BALF collected on day 3 after LPS or phosphate-buffered saline (PBS) exposure followed by PA14 infection. Significantly greater levels of myeloperoxidase (MPO) and neutrophil elastase (ELA2), both associated with neutrophil influx, were detected in the LPS + PA14 mice compared to that in the PA14 mice (Fig. 1g). In contrast, while the levels of the neutrophil chemoattractants, CXCL1 and CXCL2, in the BALF were significantly higher above baseline in the PA14 mice within 4 h after infection, this response was attenuated in mice pre-exposed to LPS, again showing a fine tuning of the inflammatory response by LPS (Fig. 1g). In agreement with the RT-qPCR data, higher levels of the cytokines IL-1β, IL-6 and IL-10 were detected in the BALF of mice infected with PA14 alone compared to those pre-exposed to LPS and then infected (Fig. 1g). However, IFN-γ protein level was higher in the BALF of the LPS + PA14 mice compared to that in the PA14 mice, also consistent with the RT-qPCR data (Fig. 1g). Histological assessment of the lung sections of the LPS + PA14 mice revealed that an expected inflammatory response was detectable in the lungs on day 3 but the inflammation resolved 3 days later (Supplementary Fig. 1a) consistent with the normal behavior of these mice followed up to 14 days after PA14 infection (Fig. 1d).

### Increase in neutrophils and interstitial macrophages in LPS-exposed lungs

We analyzed single cell suspensions of lung tissue from PA14 and LPS ± PA14 mice by multi-color flow cytometry using an optimized gating strategy (Supplementary Fig. 1b). In these and subsequent experiments, we focused on the host response to infection on day 3 after LPS treatment when the lung bacterial load was the least as compared to that in mice without LPS pre-exposure (Fig. 1b). In the 3 groups of mice-PA14, LPS and LPS + PA14, total lung cell counts significantly increased compared to that in naïve mice with a similar profile observed for CD11b+ cells (Fig. 2a). However, we detected a loss of SiglecF+CD11b$^{lo/-}$ alveolar macrophages (AMs) in these groups from numbers present in naïve mice (Fig. 2a, b). Loss of AMs has been reported in many settings of lung infection and is referred to as the macrophage disappearing reaction[16–18]. Compared to PA14 mice, lungs of both LPS only and LPS + PA14 mice showed significantly higher numbers of CD11b+Ly6G+ neutrophils and CD11b+CD64+CD24− interstitial macrophages (IMs) were detected, both cell types having high phagocytic functions. Neutrophil numbers were only slightly higher in the LPS + PA14 mice compared to that in LPS only mice (Fig. 2a). This profile also matched the lung histology profiles (Supplementary Fig. 1a). In contrast, the numbers of CD11b+Ly6C$^{hi}$CD43− classical monocytes (cMos), did not appreciably increase in response to either LPS or PA14 (Fig. 2a, b). The flow cytometry data also revealed significantly lower numbers of IMs in the PA14 mice as compared to that in the LPS-exposed mice (Fig. 2a, b). Taken together, our data suggested that although a robust neutrophilic response was mounted in both the LPS + PA14 and PA14 mice, given the divergence in the outcome in the two groups, functionally the neutrophils in the LPS + PA14 mice were more adept in bacterial clearance compared to those in the PA14 mice (Fig. 1b). Another feature of LPS pre-exposure was the significant increase in Ly6G+CD14+IFN-γ+ neutrophils as compared to a very small percentage detected in the PA14 group (Fig. 2c).

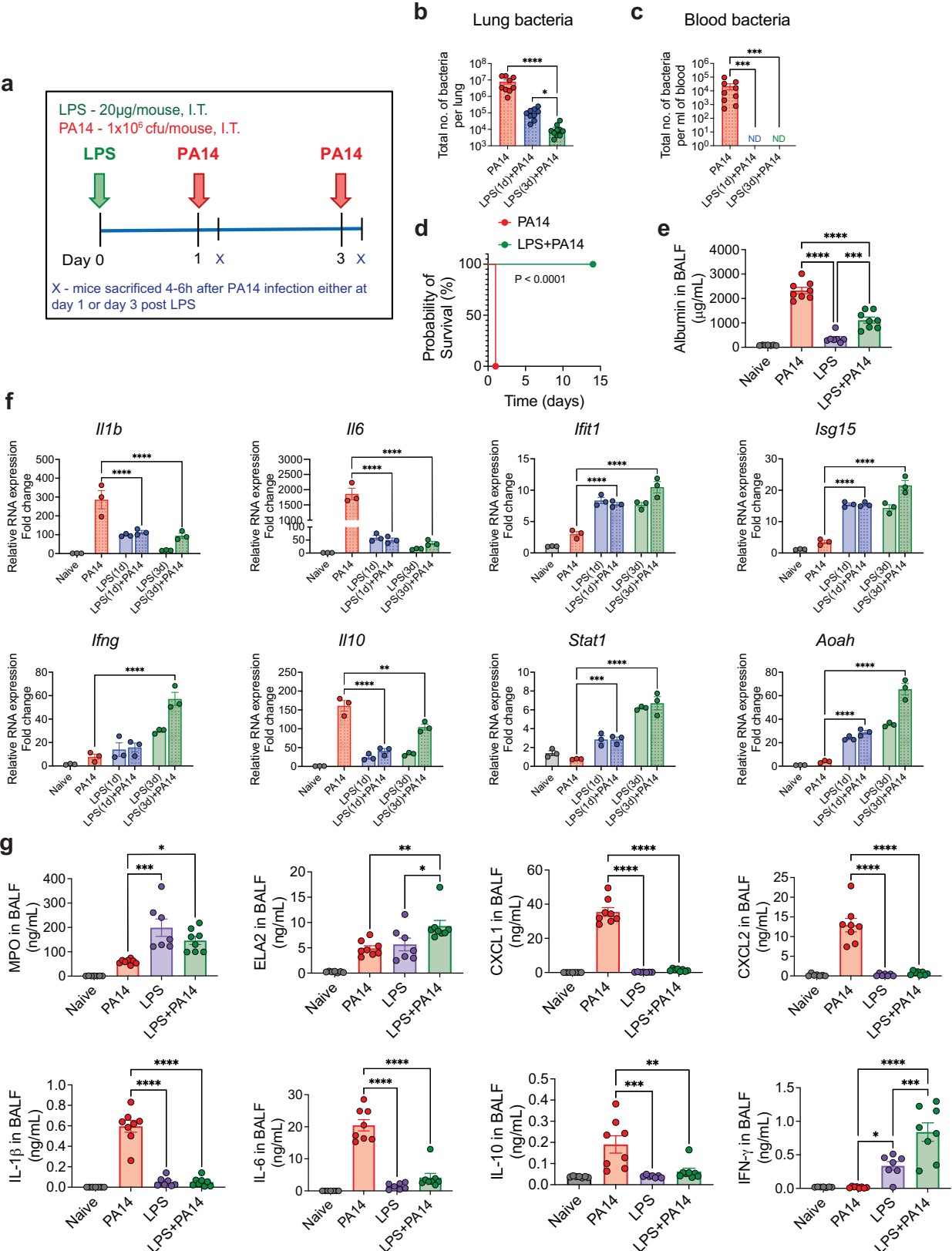

### Pre-exposure to LPS in PA-infected mice promotes specific neutrophil and IM populations as revealed by scRNA-seq

Next, for an in-depth analysis of the immune cells in the LPS + PA14 and PA14 groups, we took the approach of single cell RNA-sequencing (scRNA-seq). As performed for flow cytometry analysis (Fig. 2a–c and Supplementary Fig. 1b), mice received i.t. LPS or PBS, followed in 3 days

with PA14 via the same route. Mice were sacrificed at 4 h post infection to harvest CD11b⁺ immune cells from the lungs for scRNA-seq analysis. We focused on CD11b⁺ cells based on our flow cytometry data that showed depletion of CD11b^lo/− AMs, with concomitant increase in neutrophils and IMs, these cell types and other innate immune cells being largely CD11b⁺. Here, we implemented the Seurat based

**Fig. 1 | Pre-exposure to LPS protects mice from a lethal infection by PA14 with reduced bacterial burden and a regulated inflammatory response. a** Schematic of PA14 infection with or without LPS pre-treatment. Bacterial load showing reduced bacterial burden in the lung (**b**) and decreased peripheral dissemination (**c**) in LPS + PA14 mice compared to PA14 mice (4 h post-infection). *n* = 9 (PA14), 9 (LPS(1d)+PA14), and 10 (LPS(3d)+PA14) mice. Data were log transformed using log base 10 to adjust for differences in standard deviation prior to analysis. Data pooled from 3 independent experiments and analyzed using one-way ANOVA with the Dunn's test. **d** Kaplan-Meier survival curve showing 100% mortality of PA14 infected mice within 18–20 h of infection, whereas 100% survival of LPS + PA14 group as monitored up to 14 days post infection. *n* = 8 (PA14 and LPS + PA14) mice per group. Data representative of 3 independent experiments and analyzed using Log-rank test. **e** Higher albumin levels in BALF of PA14 mice compared to LPS ± PA14 mice (mice pre-exposed to LPS for 3 days), BALF being collected 4 h post-infection from both groups of mice and also from naïve mice and mice only treated with LPS. *n* = 7 (Naïve and LPS), and 8 (PA14 and LPS + PA14) mice. Data pooled from 2 independent experiments and analyzed using ordinary one-way ANOVA with the Tukey post-hoc test. **f** RT-qPCR analysis of gene expression of *Il1b, Il6, Ifit1, Isg15, Ifng, Il10, Stat1* and *Aoah* in the lung tissue of the four groups of mice. *n* = 3 mice per group. Data representative of 3 independent experiments and analyzed using ordinary one-way ANOVA with the Dunnett's test. **g** MPO and ELA2 protein levels associated with neutrophil influx and chemokine and cytokine levels in the BALF of the two groups of mice. *n* = 7 (Naïve and LPS), and 8 (PA14 and LPS + PA14) mice. Data pooled from 2 independent experiments and analyzed using ordinary one-way ANOVA with the Tukey post-hoc test. All data are presented as mean ± s.e.m. \**P* < 0.05, \*\**P* < 0.01, \*\*\**P* < 0.001, \*\*\*\**P* < 0.0001. Source data and exact *P* values are provided as a Source Data file.

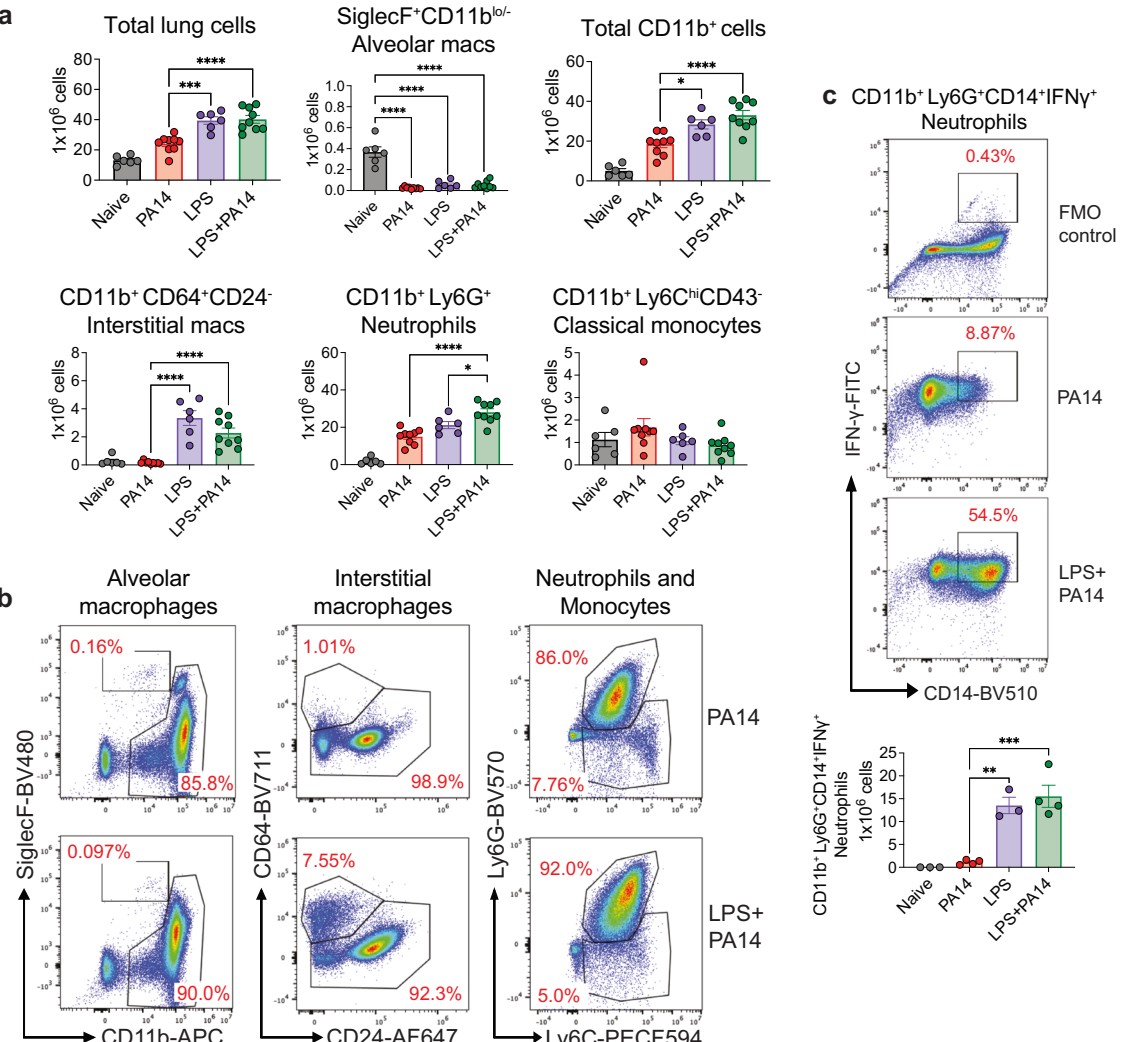

**Fig. 2 | Pre-exposure to LPS promotes an increase in neutrophils and interstitial macrophages in mouse lungs. a** Flow cytometry analysis showing total numbers of lung cells, AMs, CD11b+ cells, neutrophils, classical monocytes, and IMs. *n* = 6 (Naïve and LPS), and 9 (PA14 and LPS + PA14) mice. Data pooled from 2 independent experiments and analyzed using ordinary one-way ANOVA with the Tukey post-hoc test. **b** Representative flow plots for identification of the cell types in the lungs of the two groups of mice. **c** Representative flow plots and total counts of IFN-γ-expressing neutrophils in the lungs of the two groups of mice from 2 independent experiments. *n* = 3 (Naïve and LPS), and 4 (PA14 and LPS + PA14) mice. Data analyzed using ordinary one-way ANOVA with the Tukey post-hoc test. All data are presented as mean ± s.e.m. \**P* < 0.05, \*\**P* < 0.01, \*\*\**P* < 0.001, \*\*\*\**P* < 0.0001. Source data and exact *P* values are provided as a Source Data file.

integrative data analysis pipeline to analyze the single cell data from LPS + PA14 and PA14 samples. After additional quality filtering, we profiled 46,826 cells from LPS + PA14 and PA14 samples. We identified 14 discrete cell types based on distinct markers and after merging similar clusters noted for IM2 (4 and 5), N3 (7 and 8) and N4 (0 and 1)

(Fig. 3a, b and Supplementary Fig. 2a−c). The three biological replicates from the same condition (i.e., PA14 vs. LPS + PA14) were merged using the Seurat function merge where each condition is in the form of a combined data matrix but we can still identify which replicate the cells come from. UMAP and tSNE plots are standard representations to

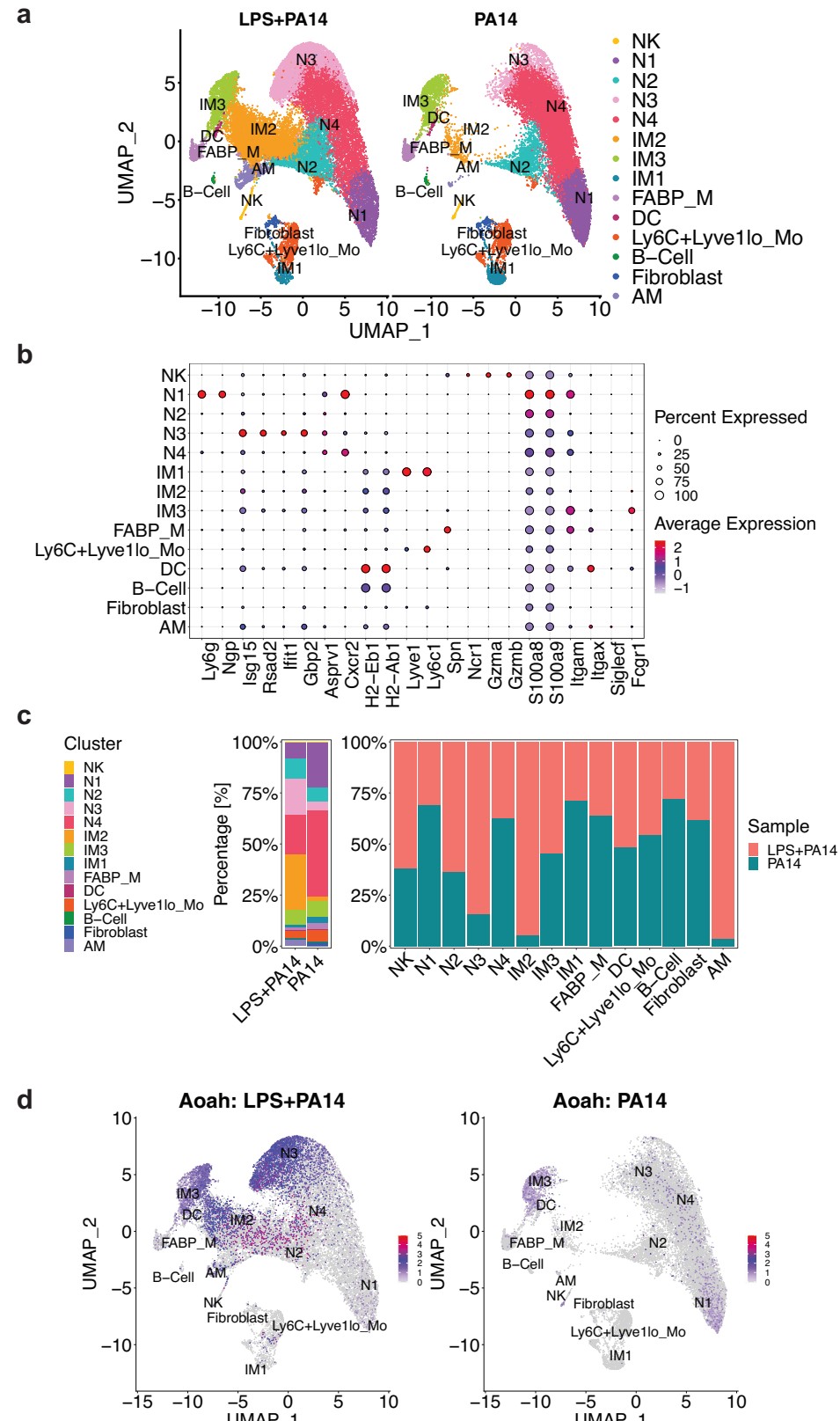

**Fig. 3 | Identification of immune cell populations in the lungs of LPS + PA14 and PA14 mice by scRNA-seq. a** UMAP embedding of the integrated expression profile of 46,826 cells from two conditions (LPS + PA14 and PA14), each comprised of three biological replicates. Distinct clusters are annotated and color-coded. **b** Expression of canonical markers used to annotate clusters, in conjunction with automated annotation using SingleR v2.2.0 (ImmGen reference scRNA-seq dataset). **c** Proportions of cell types represented in the two conditions, including the total count of cells in each cell type and the experimental condition. **d** Feature plot visualizing the expression of *Aoah*, the color of points (cells) represents the expression level of *Aoah*. Cell types are labeled, and the plot is split by experimental condition to highlight differences in expression levels between cells from LPS + PA14 and PA14 mice. *n* = 3 mice per group.

depict how well the replicates overlap each other (Supplementary Fig. 3). Manually curated annotations were further substantiated with automated annotations drawn from Immune Genome (ImmGen) database[19] using the SingleR package. The major cell clusters identified included four neutrophil populations, three IM populations, monocytes (Ly6C+Lyve1lo), mixed with smaller clusters of natural killer (NK) cells, dendritic cells (DCs), FABP+ macrophages (FABP_M) and B cells. While cell counts corresponding to most of the clusters were largely similar in the two groups, cell counts for two, a neutrophil population that we named N3, and an IM population, identified as IM2, were 5.5× and 17.5× higher in cells isolated from LPS + PA14 mice compared to those from PA14 mice (Fig. 3a and Supplementary Table 1). Although we used enriched CD11b+ cells for scRNA-seq analysis, minor populations of CD11b− cells were identified as contaminants that included AMs and fibroblasts. Neutrophils were identified using a combination of marker genes. Expression of *Asprv1*, which was characterized as a neutrophil-specific marker in both mice and humans[20–22], was useful to identify four neutrophil clusters N1-N4 (Fig. 3b, and Supplementary Fig. 2a, c). The expression of *Ly6g*, most commonly associated with mature neutrophils[23], was variable in the neutrophil populations, as also observed in other studies[24]. Clusters N1-N4 also expressed *Cxcr2*, a key chemokine receptor expressed by both mouse and human neutrophils, and the S100 genes *S100a8* and *S100a9*, also commonly expressed by neutrophils. N1 displayed features of a mature neutrophil population with high level of expression of *Ly6g* and *Ngp*. N3, which was much more abundant in the LPS + PA14 group, expressed higher levels of multiple ISGs compared to the other neutrophil clusters. N3 also displayed expression of major histocompatibility complex (MHC) genes. The three IM populations were distinguished based on differential expressions of the MHC genes, *H2-Eb1* and *H2-Ab1*, and *Lyve1*, as described previously[25–27]. Although IM1s are described as Ly6C−, the IM1s in our analysis expressed Ly6C, as was also noted in a study of pulmonary hypertension[28]. Conversely, having detected Lyve1 expression in a small fraction of the monocytes (Fig. 3b), we labeled the cells Ly6C+Lyve1lo. Figure 3c depicts the proportions of these cell clusters in the two groups of mice. The scRNA-seq data helped to localize the increased *Aoah* expression detected in the LPS + PA14 group in whole lung tissue (Fig. 1f), to N3 and two of the IM clusters, IM2 and IM3 (Fig. 3d).

## N3 and IM populations are distinguished by signatures of phagocytosis and cell killing

Differential expression analysis using the pseudobulk approach across aggregated cell clusters in the LPS + PA14 versus PA14 group was implemented using DESeq2. This analysis revealed increased expression of known LPS-inducible genes in the LPS + PA14 group, which included *Saa3*, an opsonin[29], complement factor B (*Cfb*), which activates the alternate complement pathway and plays an important role in antibacterial defense[30] and *Aw112010*, important in host defense and a suppressor of IL-10 gene expression[31] (Fig. 4a). Upregulation of *Aw112010* in the LPS + PA14 group was concordant with lower BAL IL-10 protein levels (Fig. 1g). Based on the differential expression analysis using single cell data, we examined DEGs across cell clusters in the two groups focusing on those clusters with the ability to phagocytose bacteria, which included the neutrophils, IMs and Ly6C+Lyve1loMos. As depicted in the volcano plots, some genes were upregulated in multiple clusters which included *Saa3*, *Aw112010*, the IFN-inducible gene, *Gbp2*, associated with bacterial killing[32], and the chemokines *Cxcl9* and *Cxcl10*, which in addition to their chemoattractant properties have potent antimicrobial functions[33,34] (Supplementary Fig. 4). We examined the relative expression of selected genes important in bacterial clearance in all cell clusters in the two groups of mice, LPS + PA14 and PA14, presented as violin plots. These included genes related to opsonization, generation of reactive oxygen species (ROS) via the NADPH oxidase 2 (Nox) system, which is a central player in innate immunity effecting bacterial killing in neutrophils and macrophages[35], *Gbp2*, *Cxcl9*, and *Cxcl10*. Expression of the genes, *Saa3* and *Cfb*, was only detected in the LPS + PA14 group, the major signals being in the N3, IM2, and IM3 clusters (Fig. 4b). The expression profile of the Nox subunit genes was largely similar in the two groups. However, *Cyba*, which encodes p22phox, and associates with Cybb/gp91phox forming the membrane-bound subunits of the NADPH oxidase system[35], was expressed at a higher level in N3 and the IM2 and IM3 populations in the LPS + PA14 mice relative to the PA14 mice. *Gbp2* was robustly expressed in the LPS + PA14 mice, most prominently in the N3 and IM3 clusters, N2 also displaying some expression. Both *Cxcl9* and *Cxcl10* were found to be expressed more prominently in the LPS + PA14 mice, N3, IM2, and IM3 displaying the strongest signals among the neutrophil and IM populations, *Cxcl10* being also expressed in the N2 cluster. Interestingly, although IFN-γ protein was detected by flow cytometry in the lung neutrophils, *Ifng* transcript was not detectable in any neutrophil population suggesting post-transcriptional regulation of IFN-γ production from very low levels of Ifng mRNA[36].

To gain more biological insights underlying the N3, IM2 and IM3 cell types, we used Gene Set Enrichment Analysis (GSEA) to compare pathways activated/suppressed between LPS + PA14 mice versus the PA14 mice. We observed enrichment of various pathways in the LPS + PA14 mice that included response to both Type I and Type II IFN, response to bacteria together with pathways enriched in genes related to phagocytic vesicle/phagocytic vesicle membrane and cell killing (Supplementary Fig. 5). We further compared select pathways across all cell clusters, and the ones related to response to Type I and II IFNs and to bacteria were enriched in all cell clusters (Fig. 4c). However, the pathways related to cell killing and phagocytic vesicle showed selective enrichment in clusters N3, IM2, IM3 and Ly6C+Lyve1loMos. Also, only N3, IM2 and Ly6C+Lyve1loMos showed association with the GO term hydrogen peroxide metabolic process, which comprises genes such as *Prdx5* and a superoxide dismutase (*Sod*) gene, Sod2, involved in reduction of oxidative burden in cells (Supplementary Data 1 includes GSEA data for all cell clusters). Although the neutrophil cluster N4 was more abundant in the PA14 group, it was enriched in genes associated with phagocytic vesicles but not cell killing. N2 showed a signal for cell killing but not phagocytic vesicle formation. None of these pathways was discernible in the mature neutrophil population N1 which was more numerous in the PA14 mice. The gene set cell killing in the cell cluster N3 included the genes *Cxcl9, Cxcl10*, and *Lgals3*, which encodes galectin-3 (Table S2). Prior studies have reported anti-bacterial functions of galectin-3 that include protection by galectin-3-expressing neutrophils from pneumococcal pneumonia[37] and demonstration of direct bactericidal activity of galectin-3[38]. Cell-intrinsic anti-bacterial function of galectin-3 was also described with its ability to direct the antimicrobial guanylate-binding protein, *Gbp2*, to pathogen-containing vesicles in macrophages[37–39]. Although mouse neutrophils are reported to express very low levels of galectin-3, unlike macrophages which constitute a rich source of the protein[37], Fig. 4d shows *Lgals3* expression in all neutrophil clusters and also in IM2, IM3, and the contaminating AMs in both groups of mice. However, with the greater abundance of N3 and IM2 in the LPS + PA14 mice compared to the PA14 mice, the former would be expected to produce higher levels of this antibacterial protein. Chord plot representation of the ontology terms shows the core genes enriched in these pathways and are involved in bacterial clearance via phagocytosis and killing mechanisms (Fig. 4e). For example, the phagocytic vesicle and phagocytic vesicle membrane GO terms include multiple genes related to antigen presentation, the gene *Clec4e*, known to promote phagocytosis[40] and *Cybb*, the membrane-bound partner of *Cyba* (Fig. 4b). All of the cell clusters N3, IM2, IM3, and Ly6C+Lyve1loMos displayed the chemokine genes *Cxcl10* and/or *Cxcl9* associated with cell killing consistent with the known potent anti-bacterial properties of these chemokines[33,34].

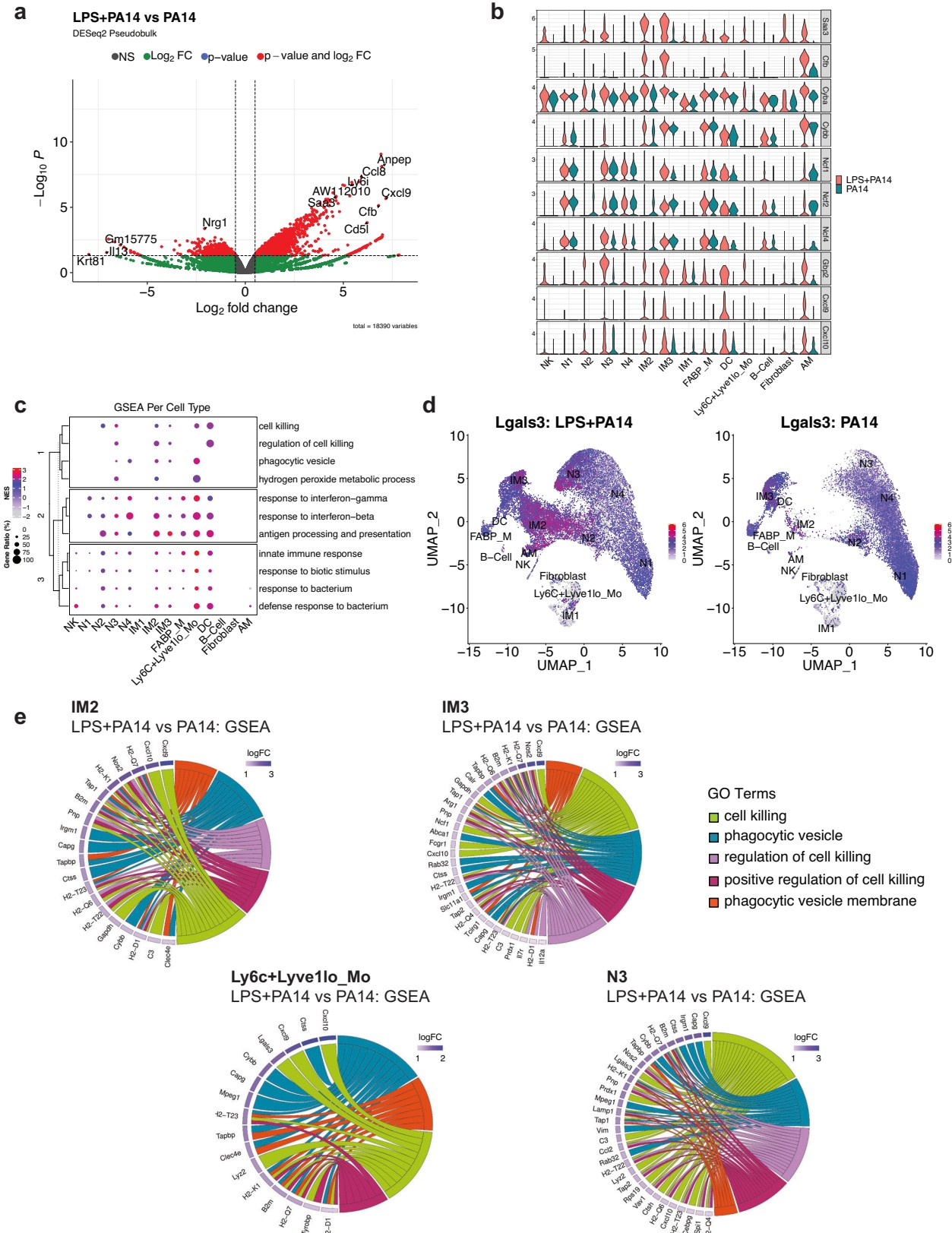

### Pre-treatment with LPS in PA14-infected mice helps to educate neutrophils for phagocytosis and cell killing

Given that the scRNA-seq data suggested increased bacterial phagocytosis and killing in mice pre-exposed to LPS, we next performed functional experiments to validate the data. We focused on neutrophils for the phagocytosis experiments given their essential role in defense against *P. aeruginosa*[12] and the impact of LPS in augmenting accumulation of neutrophils. To study phagocytosis, we performed intravital imaging using a reporter-GFP bacterium, PA-GFP, that we found elicited a similar outcome in mice as compared to PA14 with respect to physical appearance of the mice after infection, ensuing bacterial burden, and lung immune cell profile

**Fig. 4 | Differential gene expression in cell clusters in LPS + PA14 versus PA14 mice. a** Volcano plot of genes differentially expressed genes in the DESeq2-based pseudo-bulk analysis of LPS + PA14 versus PA14 cells. Red points indicate significance in both fold-change (log2FC > 0.5) and Benjamini-Hochberg (BH) adjusted $p$-value ($p < 0.05$). **b** Stacked violin plots highlighting select genes that were found to be differentially expressed between the two conditions (LPS + PA14 versus PA14). **c** A dotplot comparing the expression levels of core genes of select pathways of interest enriched in LPS + PA14 against PA14 cells. The size of the dot indicates the gene ratio, and the color of the dot shows the normalized enrichment score (NES). The dendrogram in the y-axis represents the clustering of the pathways by the NES. **d** Feature plot visualizing expression of *Lgals3*, and the color represents the average expression level across cell types. **e** Chord diagrams showing the pathways of interest from the GSEA. In each diagram, enriched clusters are shown on the right and genes contributing to the enrichment are shown on the left. Genes are colored by log fold-change value, and the color of the chords represent distinct pathway/ontology Terms.

(Supplementary Fig. 6a, b). To assess phagocytosis of PA-GFP by neutrophils, mice were treated i.t. with either LPS or PBS followed in 3 days i.t. with PA-GFP. Quantitative fluorescence intravital lung microscopy (qFILM) was performed to assess phagocytosis in the lungs of live mice at 3–4 h post-infection. Neutrophils (red) were visualized in vivo using Pacific Blue (PB)-conjugated anti-Ly6G antibody (pseudocolored red) administered intravenously (i.v.) and the microcirculation was visualized by i.v. injection of Texas Red Dextran, pseudo-colored purple. As shown in Fig. 5a–e, and in Supplementary information, included as Supplementary movies 1 and 2, LPS treatment promoted phagocytosis of PA-GFP by the neutrophils, as evidenced by increased colocalization of PB and PA-GFP in the neutrophils (Fig. 5d). Also, the percentage of PB-eGFP colocalization was significantly higher in the LPS pre-exposed mice showing that colocalization was independent from the number of neutrophils per field of view (Fig. 5e). As observed in other experiments (Fig. 2b), neutrophil counts in the lungs of the LPS-treated mice were higher than those in the PA-GFP alone group (Fig. 5c). We examined expression of a gene set that was previously described to promote the phagocytotic function of neutrophils in all cell clusters[41]. The gene, *Fcer1g*, which encodes the Fcγ receptor, and plays an important role in phagocytosis and innate immunity[42], was enriched in the N3 and IM2 clusters in the LPS + PA14 mice compared to that in the PA14 mice (Fig. 5f). Increased *Rac1* expression in N3 was also detected in these mice, Rac1 being an important component of the plasma membrane Nox2 system in neutrophils[35]. *Abca1* and *Aif1*, associated with efferocytosis of apoptotic cells and bacterial phagocytosis respectively[43,44], were expressed at a higher level in IM3s in the LPS + PA14 mice compared to that in the PA14 mice.

We next collected BAL cells from the two groups of mice at 4 h post-infection to assess intracellular bacterial load having detected cell-killing signatures in the N3 cluster by GSEA. ~100-fold more BAL cells were recovered from the LPS + PA14 mice compared to the PA14 mice (Fig. 5g). The majority were neutrophils in both, the remaining cells being AMs (Fig. 5h-k). We have not yet interrogated the nature of the neutrophil subpopulations in the BALF. The BALF was plated for bacterial quantification while BAL cell pellets were permeabilized to assess intracellular bacterial burden. A significantly lower bacterial count was detected in the BALF of LPS + PA14 mice as compared to that in the PA14 mice, with a similar 2–3-log fold reduction in intracellular bacteria in the BAL cells of LPS + PA14 mice (Fig. 5l, m). It is important to note that since the differential bacterial counts were evident after permeabilization of the BAL cells, they reflected phagocytosed intracellular bacteria and not extracellular contaminants. Thus, while significantly higher numbers of neutrophils extravasated into the air spaces in the LPS + PA14 mice, they harbored few live bacteria. In contrast, the fewer neutrophils in the PA14 mice had a significantly higher load of live bacteria suggesting inability to efficiently kill the phagocytosed pathogen. Based on the GSEA data, we also assayed galectin-3 protein levels in the BAL cells and detected significantly higher expression associated with the cells from the LPS + PA14 group (Fig. 5n). Taken together, these results provided a functional correlate to the scRNA-seq data suggesting that neutrophils played a key role in phagocytosis and killing of bacteria in mice pre-exposed to LPS.

## Galectin-3 levels in the lower respiratory tract of critically ill patients with acute respiratory failure correlate with neutrophil biomarkers and predict survival

Respiratory infections and their sequelae can precipitate acute respiratory failure (ARF). Given *Lgals3* expression in neutrophils in the mouse data, its enrichment in the highly abundant N3 cluster in the LPS + PA14 mice associated with cell killing gene signature, and detection of significantly higher galectin-3 protein levels associated with BAL neutrophils in LPS + PA14 mice, we sought to determine whether clinically significant differences in galectin-3 levels may be present in the lower respiratory tract (LRT) of patients with ARF. We have recently demonstrated that host response biomarkers can be measured reproducibly in supernatant fluid of endotracheal aspirate (ETA) specimens obtained from mechanically ventilated patients with ARF, offering clinically valid discriminations for the type and etiology of ARF[45]. Therefore, we studied a cohort of 81 patients with severe ARF, either with acute respiratory distress syndrome (ARDS) per Berlin criteria ($n = 28$, 50% men) or at-risk for ARDS ($n = 53$, 68% men) due to direct or indirect risk factors for lung injury[46]. Detailed clinical characteristics are provided in Supplementary Table 2. In ETA supernatant fluid obtained within 48 h of intubation and subsequently cryopreserved, we assayed two key host response markers associated with neutrophils that emerged from the murine experiments: galectin-3 and neutrophil elastase. From available plasma biomarker values for interleukin-(IL)-6, soluble tumor necrosis factor receptor-1 (sTNFR1), and serum bicarbonate, we classified patients into a hypo vs. hyper-inflammatory subphenotype with predicted probabilities from parsimonious logistic regression model using the Youden index cutoff of 0.274, as previously described[47,48] (Supplementary Fig. 7a). Apart from the two plasma biomarkers (IL-6 and sTNFR1), and clinical variable CO2 (serum bicarbonate), hyperinflammatory patients had statistically significant higher levels of IL-8, procalcitonin, RAGE, ST2, creatinine and white blood cell count levels, as well as a higher total respiratory rate at baseline compared to hypoinflammatory patients (all $p < 0.05$) (Supplementary Fig. 7b, c). There was no significant difference in LRT galectin-3 levels between patients with ARDS (median $n = 146.7$, interquartile range [83.4-613.1] ng/ml) vs. at-risk for ARDS (119.1 [71.3–193.1], $p = 0.25$), as well as no differences between patients with direct vs. indirect risk factors for lung injury. Among 65 patients with direct lung injury risk factors (pneumonia: 56; macro-aspiration: 7; inhalational injury: 4; pulmonary vasculitis: 1), we found no systematic difference of LRT galectin-3 levels when patients were stratified by the organisms isolated in clinical microbiologic cultures of LRT biospecimens (ETA or BAL) obtained as part of the diagnostic workup by the treating clinicians (Supplementary Fig. 8a). However, presence of neutrophils in Gram Stain examination of these clinical LRT biospecimens was strongly associated with higher galectin-3 levels in ETA research biospecimens (Supplementary Fig. 8b). Thus, LRT galectin-3 levels in our research biospecimens related to neutrophils and not specific organisms in clinical LRT biospecimens. Importantly, this observation also aligns well with prior studies that have associated high neutrophil numbers in bronchoalveolar lavage (BAL) of patients with bacterial infection[49–51]. We then examined for relationship of LRT galectin-3 with 30-day survival and host-response subphenotypes. Among all patients

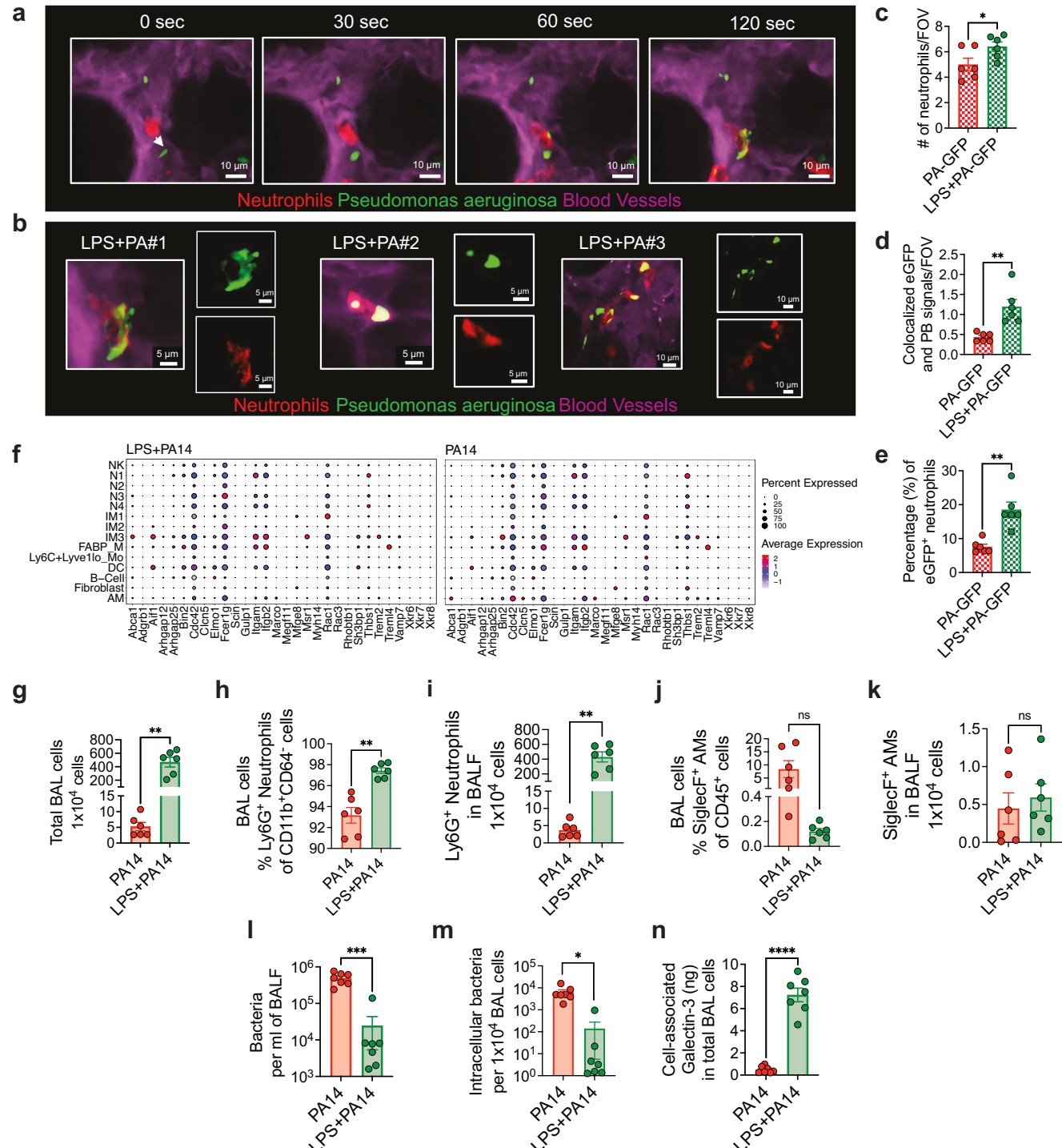

**Fig. 5 | Pre-treatment with LPS educates neutrophils for efficient phagocytosis and cell killing. a** Quantitative fluorescence intravital lung microscopy (qFILM) images of the same field of view (FOV) in the lung of an LPS + PA-GFP mouse at four different time points showing neutrophils crawling intravascularly (direction shown by white arrow) towards PA-GFP and finally phagocytosis of PA-GFP. Complete time series for (**a**) shown in Supplementary movies 1 and 2. **b** Magnified images showing phagocytosis in three different mice using merged channel pictures and isolated red and green channels. **c** Relative neutrophil counts in FOV in the two groups of mice. **d** qFILM data were analyzed as described in Methods. Phagocytosis of PA by neutrophils is shown as the average colocalization of eGFP and Pacific Blue (PB) signals within the FOV. **e** The percentage of eGFP⁺ neutrophils combining all fields of view. Representative images (**a**, **b**) and pooled data (**c**–**e**) from 3 independent experiments with *n* = 2 mice per group per experiment. Data analyzed using two-tailed unpaired *t* test with Welch's correction. **f** Dot plots

showing the expression levels of phagocytosis-related genes in all cell types, separated by condition. Flow cytometry analysis of BAL cells (**g**) with neutrophils (**h**, **i**) comprising more than 90% of the BAL cells in PA14 and 98% in LPS + PA14 mice and AMs (**j**, **k**) comprising around 10% and 0.2% respectively. **g**–**k** *n* = 3 mice per group per experiment. Data pooled from 2 independent experiments and analyzed using two-tailed unpaired *t* test with Welch's correction. **l** Bacterial burden in BALF. **m** Intracellular bacterial load in BAL cells assessed by permeabilizing BAL cells. **n** Galectin-3 protein associated with BAL cells (primarily neutrophils). **l**–**n** Data pooled from 2 independent experiments with *n* = 7 mice per group. Data were log transformed using log base 10 to adjust for differences in standard deviation prior to analysis for (**l**, **m**). Data analyzed using two-tailed unpaired *t* test with Welch's correction. All data are presented as mean ± s.e.m. *$P < 0.05$, **$P < 0.01$, ***$P < 0.001$, ****$P < 0.0001$. Source data and exact *P* values are provided as a Source Data file.

**a** Galectin-3 by 30-day mortality in all patients with ARF

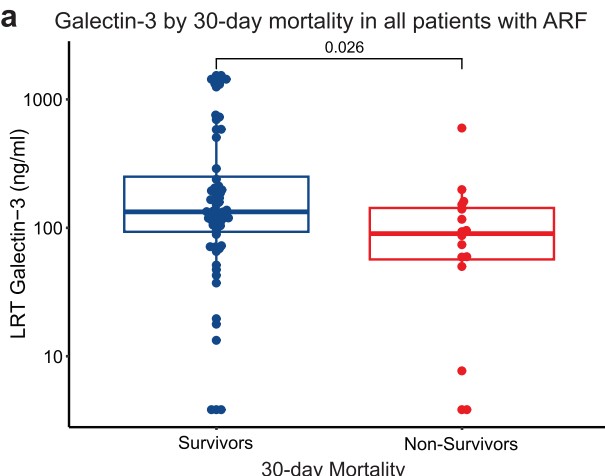

**d** Galectin-3 and Neutrophil Elastase in all patients with ARF

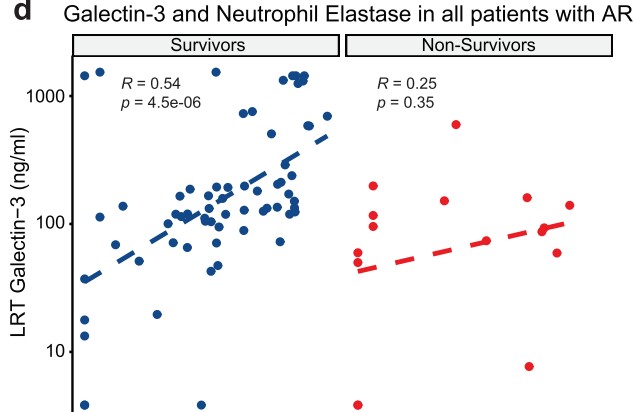

**b** Galectin-3 by 30-day mortality in all patients with ARDS

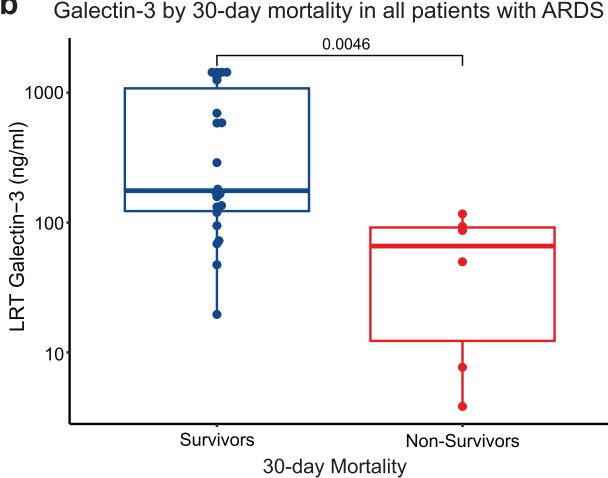

**e** Galectin-3 and Neutrophil Elastase in patients with ARDS

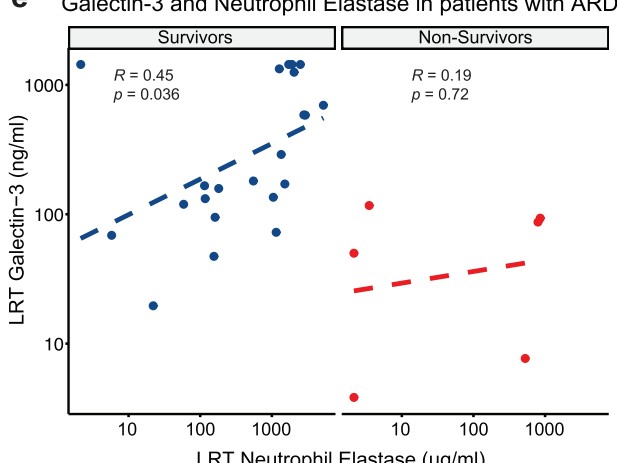

**c** Galectin-3 by Host-Response Subphenotypes in all patients with ARF

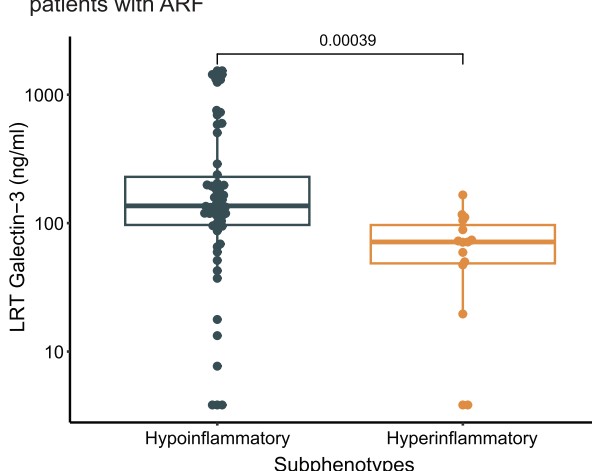

**f** Galectin-3 and Neutrophil Elastase by Host-Response Subphenotypes

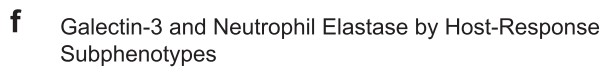

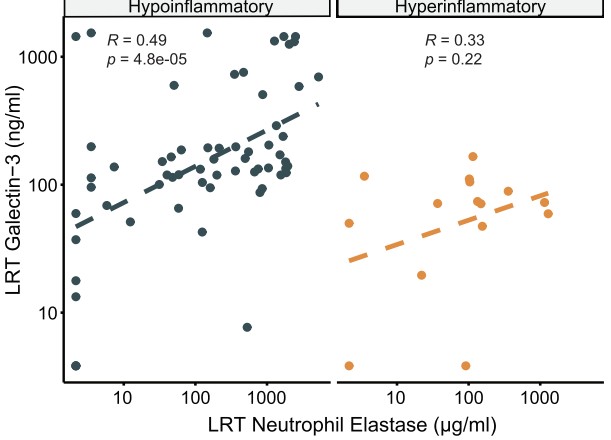

followed for 30-day survival, survivors had significantly higher LRT galectin-3 levels compared to non-survivors (Fig. 6a), and this difference was driven mostly by patients with ARDS (Fig. 6b). Similarly, we found that patients classified to the prognostically favorable hypoinflammatory subphenotype had higher LRT galectin-3 compared to patients classified to the prognostically adverse hyperinflammatory subphenotype (Fig. 6c) Stratified by 30-day survival, we found a highly

significant correlation between LRT neutrophil elastase and galectin-3 in survivors only (Fig. 6d, $r = 0.54$, $p < 0.0001$), an effect that was also driven primarily by patients with ARDS, as well as patients classified to the hypoinflammatory subphenotype (Fig. 6e–f). Given the frequent use of steroids in patients with ARF and their effects on neutrophil function, we examined for any confounding effects of steroids on the association between galectin-3 and survival and found no differences

**Fig. 6 | Galectin-3 levels in lower respiratory tract specimens predict survival and correlate with markers of neutrophil activation in patients with acute respiratory failure. a** Survivors of acute respiratory failure showing significantly higher galectin-3 levels in supernatant fluid of endotracheal aspirates (ETAs) compared to non-survivors by 30-days post intubation ($p = 0.026$). **b** ARDS survivors showing significantly higher galectin-3 levels in ETA supernatants compared to ARDS non-survivors ($p = 0.0046$). **c** Based on plasma biomarkers, patients classified as prognostically favorable hypoinflammatory subphenotype showed higher LRT galectin-3 levels compared to patients classified to the prognostically adverse hyperinflammatory subphenotype ($p = 0.00039$). **a**–**c** Data are represented as boxplots with median as the line inside the box, interquartile range (25th–75th percentile) as the box itself, whiskers extend to 1.5 times the interquartile range, and individual dots beyond whiskers signify outlier observations. *P* values from Wilcoxon test. All tests were two-sided. **d** Galectin-3 levels were significantly correlated with neutrophil elastase levels in ETA supernatants among acute respiratory failure survivors only. **e** A significant correlation between ETA levels of galectin-3 and neutrophil elastase was observed only among ARDS survivors. **f** LRT galectin-3 levels correlated with neutrophil elastase levels in ETA supernatants in hypoinflammatory subphenotype patients. *P* values from Pearson's correlation tests. $n = 81$ independent subjects. Source data is provided as a Source Data file.

between steroid and non-steroid treated patients (Supplementary Fig. 9). Overall, these human data suggested that the clinical outcome of ARF is strongly associated with increased LRT galectin-3 levels, which were significantly correlated with neutrophil elastase, measured as a marker of neutrophil accumulation.

### Increased lung bacterial load and loss of IFN-γ production from neutrophils in galectin-3 knockout mice

We next asked whether absence of galectin-3 expression reduces the LPS-mediated protective effects against *P. aeruginosa* infection. We examined whether lack of galectin-3 increased morbidity and bacterial load in murine lungs. As in all of our experiments, while both the WT and galectin-3 knockout (KO) mice infected by PA14 mice appeared sick and huddled by 4 h after infection, the galectin-3 KO LPS + PA14 mice also appeared sick and lethargic, unable to move when prompted unlike the WT LPS + PA14 mice, which did not display similar signs of morbidity. Bacterial load in the lungs of mice was 1 and 1.5 log-fold higher in the galectin-3 KO mice without or with LPS exposure respectively (Fig. 7a). However, while bacterial clearance was significantly impaired in the LPS pre-exposed galectin-3 KO mice with the mice appearing sick and huddled early after infection, they subsequently recovered after 24 h resuming normal activity. Although the galectin-3 KO mice did not show any appreciable change in the immune cell populations (Fig. 7b–d), unexpectedly, the prominent IFN-γ production from CD14$^+$ neutrophils in WT LPS + PA14 mice was almost completely lost in the absence of galectin-3 (Fig. 7e, f). This shows a previously unrecognized ability of galectin-3 to promote IFN-γ from neutrophils although whether this is a direct or indirect effect remains to be determined. Although bacterial clearance was partially compromised in the absence of galectin-3, the profiles of the immune cells were not greatly altered (Fig. 7b–d). Since our data suggested that neutrophils play an important role in bacterial clearance, we further examined whether expression of galectin-3 in neutrophils was critical for this function. We isolated BAL cells from PA14-infected WT and galectin-3 KO mice pre-exposed to LPS to assess intracellular bacterial burden. Of note, as shown in Fig. 5h, BAL cells in WT mice were primarily composed of neutrophils and it was the same in the galectin-3 KO mice. Although the total BAL cell yield from both groups was comparable (Fig. 7g), the intracellular bacterial count in BAL cells (primarily neutrophils) from the galectin-3 KO mice was approx. 1-log higher compared to that from WT cells (Fig. 7h). Collectively, these data suggest that lack of galectin-3 in neutrophils impairs early clearance of the bacteria from neutrophils which also relates to the morbidity displayed by these mice at this time. We next sought to determine whether adoptive transfer of galectin-3-expressing neutrophils isolated from the BAL fluid of LPS- treated WT mice could reduce the lung bacterial load in the LPS-treated galectin-3 KO recipients. Indeed, upon transfer of the cells, the lung bacterial burden in the recipients was significantly lower compared to that in mice that did not receive any neutrophils from WT LPS + PA14 mice (Fig. 7i). We examined the lung cells of the recipients for presence of the adoptively transferred cells. As shown in Supplementary Fig. 10a, the CTB-labeled cells comprised 3–4% of lung neutrophils in the mouse lungs. Thus, although neutrophils are short-lived cells, of the $4 \times 10^6$ WT

neutrophils transferred, ~0.3–1 × 10$^6$ survived and populated the lungs of galectin-3 KO mice (Supplementary Fig. 10a), which potentially helped to lower their bacterial burden (Fig. 7i). Taken together, our data show that galectin-3 production by neutrophils has an important role in early bacterial clearance which may have considerable significance in critically ill mechanically ventilated hospitalized patients. Our assessment of expression of a few genes associated with host defense in the lungs of WT and galectin-3 KO LPS + PA14 mice did not reveal much difference which may explain the ultimate recovery of the infected LPS-treated galectin-3 KO mice (Supplementary Fig. 10b). Only expression of *Ifng* was slightly decreased in the galectin-3 KO mice (Supplementary Fig. 10b), supporting our flow cytometry data that showed loss of IFN-γ protein in their lung neutrophils (Fig. 7e). The additional source of IFN-γ in the mice is most likely NK cells which are detectable, albeit of low abundance (Fig. 3a).

## Discussion

Although bacterial LPS can cause lung injury and shock, it can also have beneficial immunomodulatory functions. More than 65 years ago, the protective ability of LPS against respiratory infections by Gram-negative bacteria was reported[8]. However, it is not well understood how LPS trains the immune system to exert such protective functions. Our study provides new insights into how LPS programs and fine-tunes the innate immune system such that mice can be protected from a lethal infection by *P. aeruginosa*, a common pathogen in hospital-acquired pneumonia, including VAP[1–4]. scRNA-seq data revealed LPS-mediated training of the innate immune system to promote the emergence of two cell clusters, N3 and IM2, programmed with unique gene signatures for phagocytosis and cell killing focusing our attention on the molecule galectin-3 as a key player in the protective effect of LPS. This knowledge gained from scRNA-seq data was experimentally validated with the demonstration of increased phagocytosis and cell killing by neutrophils in vivo together with expression of galectin-3 protein associated with extravasated neutrophils in the airspaces of the mice pre-exposed to LPS. Exploring the relevance of these findings in humans, we found that galectin-3 levels in the ETAs of patients with ARF including those with a diagnosis of ARDS, were positively associated with improved 30-day survival. Deletion of galectin-3 in LPS + PA14 mice increased bacterial burden in the lungs and BAL cells, the latter largely composed of neutrophils, with loss of IFN-γ production by neutrophils.

Our data did not indicate that LPS-exposed mice simply had an overall advantage in accumulating higher numbers of neutrophils in the lungs. In contrast, the significant impact of LPS priming prior to infection was the specific and selective accumulation of the N3 population. This particular population exhibited enrichment in gene sets related to phagocytosis, cell killing, and dissipation of $H_2O_2$. Importantly, this unique combination of gene sets was not observed in any of the other three populations, namely N1, N2 and N4. The N3 neutrophil population was also distinguishable from the other three neutrophil populations by expression of multiple ISGs. In recent years, the ability to study cells at single cell resolution has revealed heterogeneity in neutrophils that include populations in different maturation stages and also activation states[52,53]. ISG-expressing neutrophils have

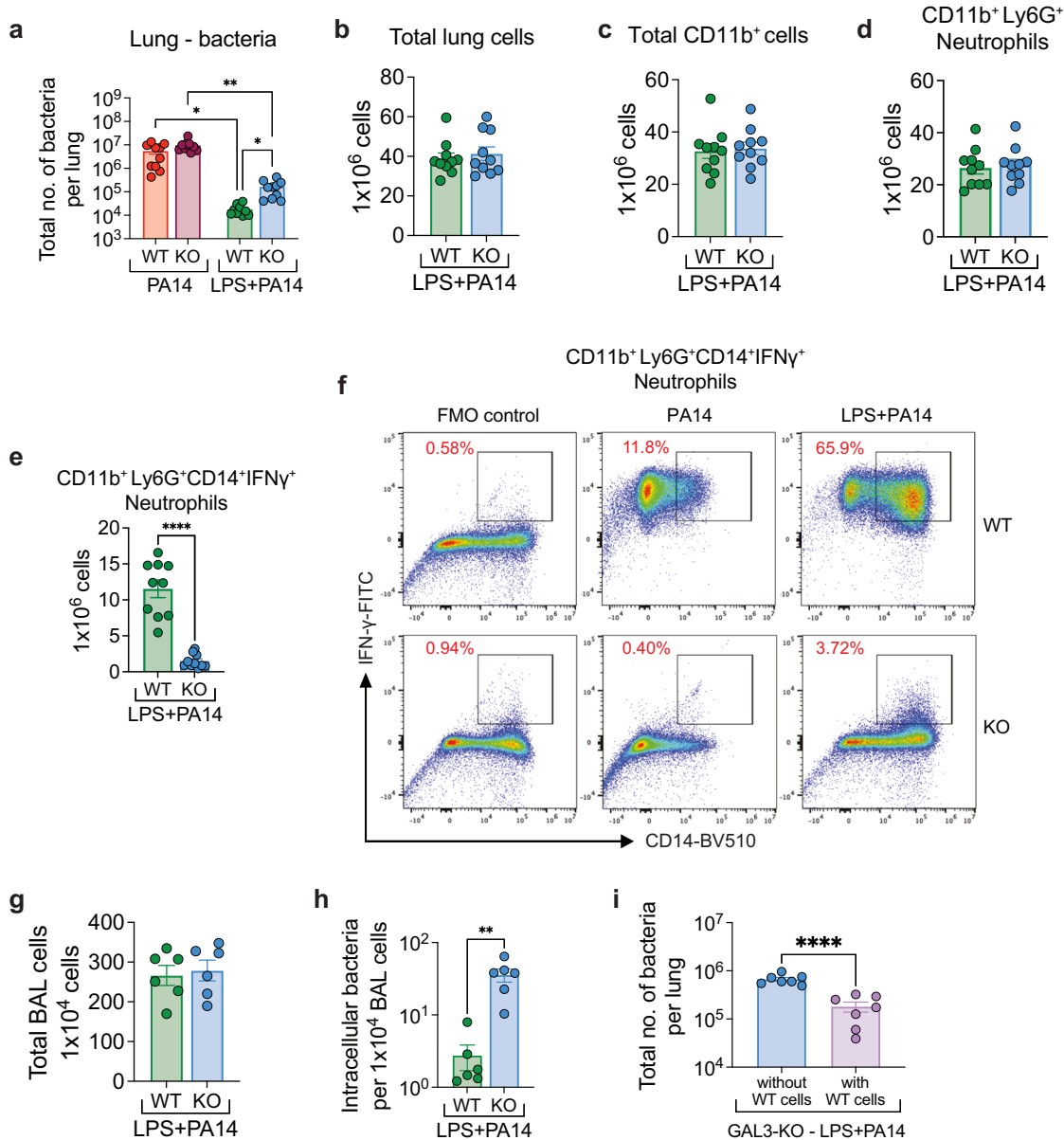

**Fig. 7 | Lack of galectin-3 in mouse increases lung bacterial burden with suppression of IFN-γ-expressing neutrophils. a** Bacterial load showing increased bacterial burden in the lungs of KO mice in both PA14 and LPS + PA14 groups as compared to the respective groups in WT mice (assayed 4 h post-infection). Data were log transformed using log base 10 to adjust for differences in standard deviation prior to analysis. Data pooled from 3 independent experiments with $n = 10$ mice per group. Data analyzed using Brown Forsythe and Welch ANOVA test with Dunnett's T3 comparisons. Flow cytometry analysis showing total numbers of lung cells (**b**), CD11b[+] cells (**c**), and neutrophils (**d**) in both WT and galectin-3 KO LPS + PA14 mice. Total cell counts (**e**) and representative flow plots (**f**) of IFN-γ-expressing neutrophils in the lungs of both WT and KO LPS + PA14 mice. **b–f** Data pooled from 3 independent experiments where $n = 10$ mice per group. **g** Flow cytometry analysis showing total BAL cells in WT and KO mice. **h** Intracellular bacterial load in BAL cells assessed by permeabilizing BAL cells from WT and KO mice. Data pooled from 2 independent experiments with $n = 3$ mice per group. **i** Bacterial burden in lungs of LPS + PA14 infected KO mice without or with adoptive transfer of BAL neutrophils isolated from LPS treated WT mice. **g–i** Data pooled from 2 independent experiments with $n = 7$ mice per group. **b–e**, **g–i** Data analyzed using two-tailed unpaired $t$ test with Welch's correction. All data are presented as mean ± s.e.m. *$P < 0.05$, **$P < 0.01$, ***$P < 0.001$, ****$P < 0.0001$. Source data and exact $P$ values are provided as a Source Data file.

been described and observed to expand during bacterial infection[53]. Although the neutrophil populations N1 and N4 were more abundant in the PA14 mice, they were clearly functionally less efficient with respect to phagocytosis and bacterial clearance, as suggested by our functional data. Unlike N3, these populations did not show co-enrichment of the gene sets associated with phagocytic vesicles and cell killing that includes the genes *Lgals3*, the chemokines *Cxcl9 and Cxcl10*[33,34], and the complement gene *C3*[54]. Not being equipped with this antibacterial armamentarium, the PA14 mice were decidedly at a disadvantage with rapid increase in bacterial loads in the lungs

accompanied by increased dissemination into the periphery causing higher mortality compared to that in the LPS-pre-exposed mice. While galectin-3 deficiency affected early bacterial clearance in LPS-treated mice with the mice appearing sick and lethargic during this time, they subsequently recovered after 24 h displaying normal activity. It is possible that the other molecules comprising the cell-killing pathway, *Cxcl9, Cxcl10,* and *C3* also aid in bacterial clearance, although galectin-3 is important for early eradication. The higher bacterial load in galectin-3-deficient BAL neutrophils compared to that in their WT counterparts and the ability of BAL neutrophils from LPS-exposed WT mice to lower

lung bacterial load in LPS-treated galectin-3 KO mice emphasizes the importance of programming of lung neutrophils in LPS-induced resistance to bacterial infection.

LPS-induced immune tolerance caused by repeated LPS exposure was recognized decades ago[55], which results in a transient suppression of the immune system to prevent tissue damage from an over-exuberant immune response[56]. LPS-induced resistance to bacterial infection clearly does not induce immune tolerance that would hamper bacterial clearance. On the contrary, we show that LPS trains the immune system to mount an effective immune response against a lethal dose of PA14. LPS-induced expression of the enzyme Aoah, first reported in neutrophils[13], reduces the toxic effects of LPS and also prevents immune tolerance by allowing a subsequent immune response[13–15,57]. Thus, increased Aoah expression in the cell clusters, N3, IM2, and IM3 in LPS + PA14 mice can be envisaged to have played an important role in the net immune response in these mice that ultimately protected them from a lethal bacterial infection. LPS-induced Aoah expression has been also shown to limit lung injury[58]. While increased inflammation induced by LPS present in the outer membrane of invading Gram negative bacteria is essential for clearance of the pathogen, this response needs to be fine-tuned to prevent collateral tissue damage. In our study, LPS-mediated conditioning of the immune system tempered inflammation in response to PA14 by suppressing IL-1β and IL-6 but also suppressed IL-10 that would have promoted immune tolerance. However, the opposite was true for IFN-γ whose expression was boosted by LPS. Increased expression of AW112010 in multiple cell populations in LPS-exposed mice may also have played a role in mounting of a regulated cytokine response in this group of mice. AW112010 was previously shown to suppress IL-10 production and promote IL-12 expression in macrophages[31], and in combination this would promote IFN-γ production, IL-12 being an inducer and IL-10 a suppressor of IFN-γ gene expression. Whether Aoah and AW112010 are functionally linked remains to be determined in future studies. Collectively, our data highlight the importance of a regulated inflammatory response in the host's resilience to lethal infection. Neutrophils and macrophages constitute the frontline of host defense being capable of pathogen phagocytosis and fitted with a range of antimicrobial molecules. An initial wave of cell recruitment to the infected tissue is critical for the prompt eradication of the invading pathogen. To achieve this goal, multiple cytokines and chemokines are rapidly produced. However, this burst in the host response needs exquisite control to permit the right mix of cells in the tissue and the inflammatory response needs to be curtailed in a timely fashion since persistent neutrophilic inflammation induces lung injury, as we previously showed[59]. The same molecules that initially help to eliminate the pathogen can also lead to deleterious consequences if their production continues unabated. A hyperinflammatory subphenotype has been associated with worse prognosis in ARDS[60].

Galectin-3 was shown to promote neutrophil extravasation into the alveolar space and to also bind to the surface of neutrophils although the neutrophils were not the source of galectin-3[61]. It is possible that among the four neutrophil populations, N3 preferentially migrated into the alveolar space resulting in 100X more neutrophil accumulation in the LPS + PA14 mice, and because of increased expression of genes involved in phagocytosis and cell killing efficiently cleared bacteria from the alveolar space. Clearance of P. aeruginosa from the alveolar space is important for survival, as suggested by studies in both mice[62] and humans, the latter involving immunohistochemical analysis of autopsy tissue of pneumonia patients[63]. It is well recognized that neutrophil extravasation into the alveolar space primes/activates neutrophils for increased superoxide production[64,65]. It is important to note, however, that while many cell types are known to express galectin-3, which includes lung macrophages and also stromal cells, mouse neutrophils are not considered a major source[61] although human neutrophils can express galectin-3[66]. Thus, although

the feature plot shows Lgals3 mRNA signal in the different neutrophil clusters, galectin-3 protein production and secretion may have been specific to the N3 population. Of note, the galectin-3 protein does not have a conventional signal sequence for secretion and its secretion mechanism is not well understood[67]. Galectin-3 also has anti-apoptotic functions in which the functional domain is highly similar to that in the best-known anti-apoptotic molecule, Bcl2[68]. Galectin-3 may thus also play a role in the survival of the N3 population allowing its accumulation over the 3-day period after LPS exposure. While it is likely that the galectin-3 protein detected in the neutrophil-enriched BAL cell pellet from the LPS + PA14 mice was produced by the extravasated neutrophils themselves, it is possible our assay detected cell-associated galectin-3 given that galectin-3 can bind to the surface of neutrophils[61]. Regardless, our data suggest that galectin-3 detected in association with the BAL neutrophils played an important role in early bacterial clearance in the LPS + PA14 mice. Notably, galectin-3 was shown to boost activation of exudated but not blood neutrophils in humans[69]. Galectin-3 may help to kill bacteria both extracellularly and intracellularly. Secreted galectin-3 can have direct bacteriostatic effects[38]. Intracellularly, galectin-3 was shown to guide the anti-bacterial protein Gbp2 to pathogen-harboring phagosomes identified by insertion of bacterial secretion systems into the pathogen-containing vesicles[39]. Galectin-3-Gbp2 interaction may be important for early bacterial eradication in both the neutrophil population N3 and the IM population IM2 showing increased Gbp2 expression. Gbp family of proteins are induced by both type I and type II IFNs and these proteins can promote oxidative killing by depositing NOX2 on bacteria-containing phagosomes[70].

IFN-γ expression by neutrophils was previously described early after infection of mice by the Gram-positive bacteria Streptococcus pneumoniae and Staphylococcus aureus but not by the Gram-negative bacteria P. aeruginosa and Escherichia coli[71]. Further, neutrophil-expressed IFN-γ was shown to enhance host defense against S. pneumoniae[72]. We too detected only a small population of IFN-γ+ neutrophils in the PA14 mice at 4 h after infection but a large increase in their number was evident upon LPS treatment over 3 days. It is possible that galectin-3 caused increased Ifng expression in the N3 neutrophils in an autocrine fashion because of the complete loss of IFN-γ production by neutrophils in galectin-3 KO mice, with N3 numbers being very small without LPS pre-exposure and enrichment of the cell killing gene set comprising Lgals3 evident only in the N3 population. IFN-γ expressed by the neutrophils would also induce Cxcl9 and Cxl10 which have well-documented antibacterial/microbicidal properties[33,34]. In addition to IFN-γ, the enhanced Type I IFN signature detected in the lungs of the LPS-exposed mice would also promote expression of these chemokines.

Our data derived from ETAs of patients with ARF suggest a protective role of high galectin-3 levels in the lungs of these patients. Importantly, the highly significant correlation between LRT neutrophil elastase and galectin-3 in survivors, but not in non-survivors of ARDS, suggests protective effects of both galectin-3 and neutrophils, as also suggested by our findings in the mice. Whether this protective effect involves an advantage in these patients in combating microbial infections via the various functions of galectin-3 remains to be determined in future studies. However, The early anti-bacterial function of galectin-3 identified in the LPS-treated mice may provide the survival advantage that we observe in critically ill ICU patients with higher galectin-3 levels in their ETAs. Alternatively, compared to other anti-bacterial proteins, in critically ill patients, galectin-3 may have a more crucial role in host defense. Host-response subphenotypes based on plasma biomarkers have shown robust prognostic value in multiple cohorts of critically ill patients[48]. Our data provide further insights into the pathobiology of these blood-based subphenotyping by highlighting that the prognostically adverse hyperinflammatory subphenotype is associated with significantly lower galectin-3 levels in the LRT. These results underscore the importance of a regulated

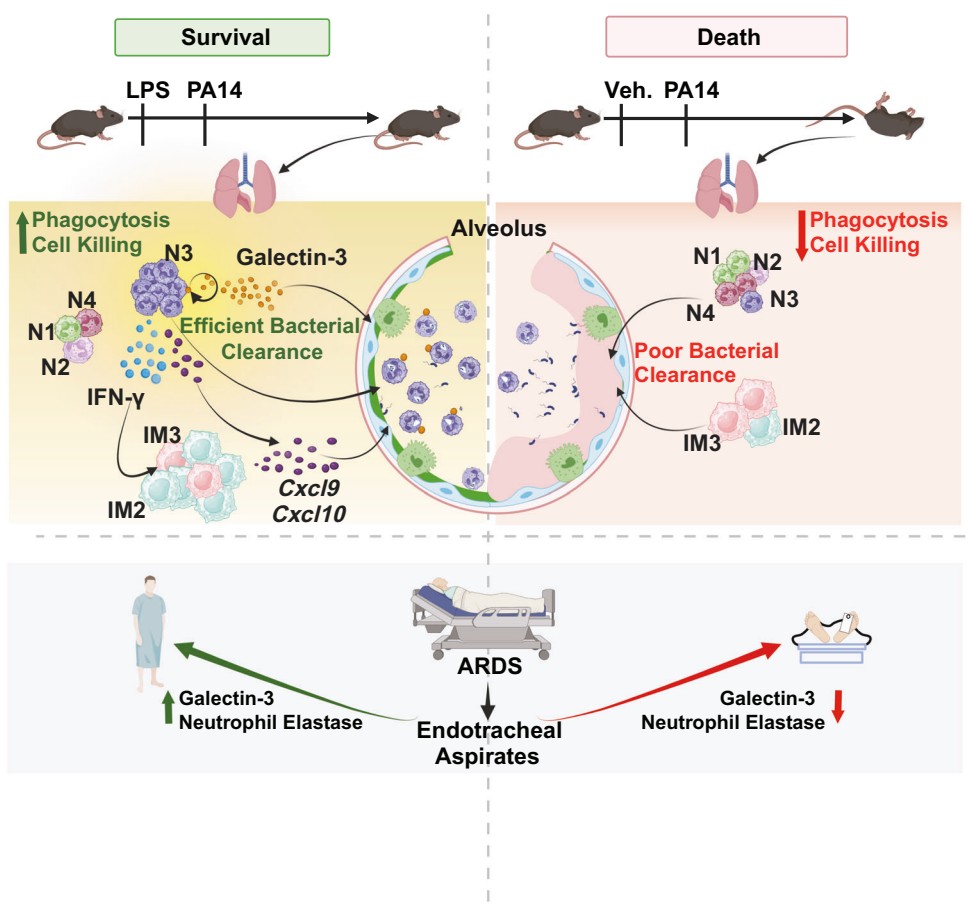

**Fig. 8 | Schematic depicting the protective role of galectin-3 in mice and humans, as revealed in our study.** A single low-dose LPS administration in the lungs of mice results in 100% survival of mice infected with a lethal dose of the virulent strain of *Pseudomonas aeruginosa*, PA14. scRNA-seq and downstream bioinformatic of lung cells of PA14-infected mice that were pre-exposed to LPS or not revealed expansion of a specific neutrophil population, we named N3, and an IM population resembling IM2 macrophages described in the literature, in the LPS pre-exposed mice. Among the four neutrophil populations, compared to the N3 population in PA14 mice, the N3 population in the LPS-treated mice was significantly more enriched in pathways associated with bacterial phagocytosis and cell killing. The cell-killing pathway comprised multiple genes with well-documented anti-bacterial functions including *Lgals3, Cxcl9, Cxcl10, and Ifng*. BAL cells were 100x more abundant in LPS-exposed mice compared to PA14 mice, in both >95% being neutrophils. BAL-neutrophil-associated levels of the anti-bacterial protein galectin-3 (encoded by *Lgals3*), were also significantly higher in the LPS-treated mice. The host-protective role of galectin-3 was also evident in critically ill hospitalized patients in the ICU with acute respiratory failure (ARF). High galectin-3 levels and a high neutrophil signature in endotracheal aspirates of patients with ARF were associated with greater survival, the data being primarily driven by patients with a diagnosis of ARDS. The schematic was created with BioRender (www.biorender.com).

inflammatory response contributing to the host's resilience against lethal infection.

The limitations in our study include lack of data on the characteristics of the neutrophil subpopulations in the airspaces of the mice and the ETAs collected from the ARF patients. We have not confirmed that such populations resemble the N3 population identified by scRNA-seq. The question of whether the absence of galectin-3 leads to increased susceptibility to infection in mice remains to be determined. It will be intriguing to explore whether modified forms of LPS with reduced toxicity compared to native LPS can exert similar protective effects. Such investigations could pave the way for prophylactic use of these molecules in ventilated patients to prevent VAP. Although we found independent associations for neutrophil biomarkers in both clinical and research LRT biospecimens with galectin-3 levels, we could not prove that measured galectin-3 was indeed expressed and secreted by neutrophils. The associations between LRT galectin-3 and survival in ARF patients are hypothesis-generating for a biologically and clinically relevant role of galectin-3 expression in the LRT of critically ill patients. In conclusion, our findings highlight lung-

protective functions of neutrophils and galectin-3 associated with increased survival in patients with ARF/ARDS (Fig. 8).

## Methods

Our research is in compliance with the guidelines of the Institutional Animal Care and Use Committee (IACUC) for mouse studies and the Institutional Review Board (IRB) for human studies at the University of Pittsburgh.

### Mice

C57BL/6J mice (Cat# 000664) and B6.Cg-Lgals3tm1Poi/J (Cat# 006338) were purchased from The Jackson Laboratory. All protocols involving animal experiments were approved by the Institutional Animal Care and Use Committee (IACUC) at the University of Pittsburgh. 10–12 week old mice were used in all experiments.

### Human samples—clinical cohort

We prospectively investigated 81 patients with ARF who were admitted to intensive care units (ICUs) at UPMC Presbyterian/Shadyside

hospitals between April 2016 and February 2020. All patients were intubated and managed with invasive mechanical ventilation for hypoxemic respiratory failure. Following admission to the ICU and obtaining informed consent from patients or their legally authorized representatives (IRB protocol STUDY19050099), we collected baseline research ETA biospecimens within 72 h from intubation. The ETA sample was collected with a standardized protocol via instillation of 5 mL of sterile 0.9% saline and retrieval of airway secretions in a closed specimen system through advancement of the in-line suction catheter and without breaking seal in the ventilatory circuit. A consensus committee of ≥3 physician-scientists certified in critical care reviewed clinical and radiographic data and performed retrospective classifications of the etiology and severity of ARF without any knowledge of experimental data or outcomes. We retrospectively classified subjects as having ARDS per Berlin criteria or being at risk for ARDS because of the presence of direct (pneumonia or aspiration) or indirect (e.g., extrapulmonary sepsis or acute pancreatitis) lung-injury risk factors although lacking ARDS diagnostic criteria. We followed patients prospectively for cumulative mortality and ventilator-free days (VFDs) at 30 days.

### Bacterial strains and growth conditions

The *Pseudomonas aeruginosa* (PA) reference strain PA14 was donated as a gift to Dr. Janet Lee by Dr. Zhenyu Cheng, formerly in Dr. Frederick Ausubel's laboratory at Harvard University. PA14 bacteria were grown overnight in Luria-Bertani (LB) broth culture medium (Fisher Scientific, Cat# BP1427-500) at 37 °C with shaking at 250 rpm. GFP-labeled reporter bacterium PA-GFP (ATCC-15692GFP) was cultured in Nutrient Broth (NB) (BD, Cat# DF0003-17-8) in the presence of 300 µg/ml ampicillin (Millipore Sigma, Cat# A5354) likewise. Optical density of the bacterial cultures was measured using a spectrophotometer (Bio-Rad Laboratories) at 600 nm wavelength to estimate the dose of infection. For colony counting, samples were plated either in LB agar (Fisher Scientific, Cat# 50-213-401) or Nutrient agar (BD, Cat# DF0001-17-0) supplemented with 300 µg/ml ampicillin for PA14 or PA-GFP respectively and grown overnight at 37 °C.

### PA infection model

Ultrapure LPS (20 µg/mouse) (InvivoGen, Cat# tlrl-3pelps) was administered intratracheally (i.t.) into mice. LPS was dissolved in LPS-free ultrapure water to prepare a stock solution and diluted further in LPS-free ultrapure PBS for working stocks. Naïve mice received the same PBS. Mice were infected with *Pseudomonas* (PA14 or PA-GFP) bacteria ($1 \times 10^6$ cfu/mouse) i.t. either on day 1 or 3 post-LPS exposure. Since we did not want any mice to die after infection before we had the opportunity to harvest the lungs and other samples, we always started our tissue harvest promptly at 4 h after infection to complete all processing within the next 2 h. Blood was collected by cardiac puncture to assay bacterial burden. Bronchoalveolar lavage fluid (BALF) was collected using a total of 3 ml of PBS (Cytiva, Cat# SH30256LS) by injecting 1 ml at a time 3 times. Cytokines were analyzed in the first 1 ml of BALF collected, while BAL cells were pooled from the total 3 ml. The lungs were harvested and homogenized in RPMI-1640 medium (Gibco, Cat# 61870-036) and used for estimation of bacterial load and RNA isolation. Mice were also monitored for survival over 14 days after PA14 infection. Of note, in compliance with our IACUC guidelines, which do not permit death as an end-point for survival studies, animals were sacrificed as soon as they fit criteria for euthanasia based on body temperature, general appearance (raised fur, withdrawn, huddled, no inclination to eat or drink) and weight loss.

### Adoptive transfer of BAL cells

BAL cells were collected from WT mice 3 days after 1 dose of LPS instillation (20 µg delivered i.t.). Cells were counted and labeled with Cell Trace Blue (CTB) (Thermo Fisher Scientific, Cat# C34568) per the

manufacturer's instructions. Briefly, BAL cells were labeled with CTB (5 µM) for 20 min at 37 °C with constant mixing. Cells were then washed with 2% FCS in PBS and $4 \times 10^6$ cells were adoptively transferred i.t. to galectin-3 KO mice. After 20 min, the mice were infected with PA14 as described above.

### Analysis of bacterial load

Bacterial burden (PA14) in mouse lung, blood, BALF, and BAL cells was estimated by counting the bacterial colonies that grew from the respective samples after plating in LB agar plates. For intracellular bacterial load, BAL cell pellets were washed 3 times with ice-cold PBS and then permeabilized using 0.05% saponin (Millipore Sigma, Cat# S7900)[73] and plated for bacterial count.

### RNA isolation and quantitative real-time Polymerase Chain Reaction (PCR)

Trizol-LS (Life Technologies, Cat# 10296028) was added to the lung homogenates and RNA was isolated using RNeasy plus mini kit (Qiagen, Cat# 74136) according to the manufacturer's instructions. cDNA was synthesized using a High-Capacity cDNA Reverse Transcription Kit (Applied Biosystems, Cat# 4368813) according to the manufacturer's instructions. Quantitative real-time PCR was performed using validated TaqMan Gene expression assays with primers and probe sets (Life Technologies) for *Il1b* (Mm00434228_m1), *Il6* (Mm0044619 0_m1), *Ifit1* (Mm00515153_m1), *Isg15* (Mm01705338_s1), *Ifng* (Mm 01168134_m1), *Il10* (Mm01288386_m1), *Stat1* (Mm01257281_m1) and *Aoah* (Mm00600104_m1), *Cxcl9* (Mm00434946_m1), *Cxcl10* (Mm00 445235_m1), and *C3* (Mm01232779_m1). Gene expression was normalized to *Hprt* (Mm03024075_m1) expression.

### ELISA and multiplex assay

Debris-free BALF was used for protein assay. ELISA kits were used for mouse albumin (Abcam, ab108792), myeloperoxidase (MPO) (R&D Systems, DY3667), neutrophil elastase (ELA2) (R&D Systems, DY4517), CXCL2 (R&D Systems, DY452) and galectin-3 (R&D Systems, DY1197). Levels of mouse IL-1β, IL-6, IL-10, IFN-γ, and CXCL1 proteins were measured using customized multiplex assay from Bio-Rad Laboratories (Bioplex Pro Mouse Cytokine Group 1). Human galectin-3 and neutrophil elastase were measured in debris-free ETA samples using human-specific galectin-3 (R&D Systems, DY1154) and neutrophil elastase (R&D Systems, DY9167) ELISA kits respectively. All assays were performed according to the respective manufacturer's protocol.

### Histology

Lungs were perfused and fixed in 2% paraformaldehyde solution. Following fixation, lungs were embedded in paraffin blocks, and 4-µm sections were mounted onto glass slides for histological analysis. Lung sections were stained with haemotoxylin (Leica, Cat# 3801571) and eosin (Leica, Cat# 3801619) solutions following an established protocol[74]. Briefly, lung sections were deparaffinized using xylene (Fisher Scientific, Cat# 9990505), then hydrated using decreasing concentrations of alcohol, and washed with distilled water before staining with haemotoxylin solution and counterstained with eosin. The sections were then dehydrated with increasing alcohol concentrations, followed by clearing with xylene and mounting on glass slides with DPX. Images were taken at ×10 magnification.

### Lung single-cell isolation

Homogenized lungs were digested using collagenase (0.7 mg/mL) (Roche, Cat# 10103578001) and DNase I (30 µg/mL) (Roche, Cat# 10104159001) in RPMI-1640 medium for 20 min at 37 °C with constant mixing and then dissociated using a gentleMACS dissociator (Miltenyi Biotech) according to the manufacturer's protocol. Single-cell suspensions were obtained by passing the dissociated tissue through a 70 µM cell strainer (Fisher Scientific, Cat# 22363548) followed by RBC

lysis using BD Pharmlyse (BD Biosciences, Cat# 555899). Isolated cells were then counted, stained with fluorochrome-conjugated antibodies, and used for flow cytometry analysis. In pilot experiments, to confirm lung residency of the depicted immune cells, we intravenously injected anti-CD45.2 antibody (Ab) (Clone: 104) (Biolegend, Cat# 109808) followed by rapid sacrifice within 3 mins and all CD45.2⁻ cells were considered lung resident[75].

## Multi-color flow cytometry
The fluorochrome-conjugated antibodies used were: anti-mouse CD45-PerCP/Cyanine5.5 (Dilution: 1:500) (Clone: 30-F11) (Biolegend, Cat# 103131), SiglecF-BV480 (Dilution: 1:500) (Clone: E50-2440) (BD Biosciences, Cat# 746668), CD11b-APC (Dilution: 1:500) (Clone: M1/70) (Biolegend, Cat# 101211), CD11c-BV785 (Dilution: 1:500) (Clone: N418) (Biolegend, Cat# 117335), CD64-BV711 (Dilution: 1:500) (Clone: X54-5/7.1) (Biolegend, Cat# 139311), CD24-AlexaFluor647(Dilution: 1:500) (Clone: M1/69) (Biolegend, Cat# 101818), Ly6G-BV570 (Dilution: 1:500) (Clone: 1A8) (Biolegend, Cat# 127629), Ly6C-PE-CF594 (Dilution: 1:500) (Clone: AL-21) (BD Biosciences, Cat# 562728), CD43-APC-R700 (Dilution: 1:500) (Clone: S7) (BD, Cat# 565532), CD14-BV510 (Dilution: 1:500) (Clone: Sa14-2) (Biolegend, Cat# 123323). Intracellular staining for cytokine was carried out using IFN-γ-FITC (Dilution: 1:500) (Clone: XMG1.2) (BD Biosciences, Cat# 554411) antibody in Foxp3 transcription factor staining buffer (eBioscience, Cat# 00-5523-00) according to manufacturer's protocol. To assess cell viability, FVD EFluor 780 (Dilution: 1:3000) (eBioscience, Cat# 65-0865-14) was used. To reduce potential non-specific antibody staining caused by IgG receptors, mouse Fc block (BD Biosciences, Cat# 553142) was used prior to antibody staining. Stained cells were examined on Cytek Aurora (Cytek Biosciences), and data were analyzed using FlowJo software (TreeStar). Alveolar macrophages were identified as SiglecF⁺CD11b^{lo/−}, interstitial macrophages as CD11b⁺CD64⁺CD24⁻, neutrophils as CD11b⁺Ly6G⁺, classical monocytes as CD11b⁺Ly6C^{hi}CD43⁻. CTB positive neutrophils accounted for adoptively transferred neutrophils in the mice. For immune cell analysis on FlowJo, live cells were gated on singlets based on live/dead staining. Total immune cells were gated based on CD45 staining and further sub-gating was performed based on above mentioned markers for specific cell populations. Fluorescence minus one (FMO) controls were used for proper gating of cell populations. Following enzymatic digestion of lung tissue, the total number of cells isolated from the lungs of each mouse was counted using Cellometer 2000 (Nexcelom), and then percentages of specific gated cell populations in flow cytometry data were used to calculate the actual cell counts.

## Magnetic labeling and purification of CD11b⁺ cells from lungs
CD11b⁺ cells from lung single-cell suspensions were purified using anti-CD11b magnetic microbeads (Miltenyi Biotec, Cat# 130-126-725) by positive selection according to the manufacturer's protocol. The enriched CD11b⁺ cells were counted and used for sequencing analyses.

## Single cell RNA sequencing (scRNA-seq) and preprocessing of data
CD11b⁺ cells were passed through a 40-μM cell strainer and enumerated by Cellometer 2000 (Nexcelom) immediately before loading onto a 10x Chromium controller for cell capture (targeting 5000 cells per sample) using the Chromium Single Cell Immune Profiling v2- Dual index kits (10x Genomics, Cat# 1000263). Gene expression libraries were constructed following protocols from 10x Genomics. Final libraries were QCed by Agilent TapeStation and then sequenced using Illumina's Novaseq 6000 targeting 50,000 reads per cell. Three biological replicates each of LPS + PA14 and PA14 samples were subjected to sequencing individually. The individual preliminary sequencing BCL files were demultiplexed and converted to FASTQ files. The 10x Genomics protocol was used (CellRanger V.4.0.0) for alignment to the

GRCm38 (mm10) mouse genome and generated the gene-cell expression matrices for the LPS + PA14 and PA14 samples. The Seurat (v 4.3.0) based workflow was implemented to pre-process the count matrices generated from the LPS + PA14 and PA14 samples. Pre-processing steps included merging of the replicates with quality control being performed at both the cell and gene levels. Quality control plots were used to assess and filter the number of counts per barcode and the number of genes per barcode. Further, we discarded cells with mitochondrial content higher than 10%. We excluded empty doublets and possible multiplets from the analysis. After QC, 46,826 high quality cells were subset for further analysis.

## Analysis of scRNA-seq data
The biological replicates for each condition (LPS + PA14, PA14), and the data from the corresponding samples were analyzed using the Seurat integrative data analysis pipeline[76,77]. The "FindVariableFeatures" function was used to select the top highly variable integration anchors which were then used as an input to the data dimensionality reduction using principal component analysis. Robust bootstrapping methods such as the jackstraw procedure and elbow heuristics were employed to identify and validate the 15 significant principal components selected as input for the Uniform Manifold Approximation and Projection (UMAP). Next, the graph-based clustering function from Seurat was used to generate the clusters with the cluster resolution set at 0.5, which was further validated using the clustree package (v0.5.0).

## Identification of cell clusters
For cell type annotation, the top marker genes were calculated for each cluster using the "FindAllMarkers" function from Seurat. Subsequently, the cells were classified into 14 cell types, based on the expression of canonical markers from relevant literature, and expert knowledge. In addition, an unbiased automated annotation was implemented using the SingleR package (v2.2.0)[78]. For automated annotation, the ImmGenData dataset from the celldex package (v1.10.0) was utilized as ref. 19.

## Identification of DEGs between LPS + PA14 and PA14 samples
Differential gene expression analysis was performed within each identified cell type across LPS + PA14 and PA14 samples using DESeq2. Differentially expressed genes were selected at a threshold of (adjusted $P < 0.05$, $|\log2\ FC| > 0.25$) based on Benjamini-Hochberg correction. Volcano plots highlighting the differentially expressed genes were generated using the EnhancedVolcano package (v1.18.0). Gene Set Enrichment Analysis (GSEA) was performed to identify potential enriched pathways in LPS + PA14 and PA14 samples. Gene set collections including the C2 (curated gene sets), and C5 (ontology gene sets) available from the Molecular Signature Database (MSigDB, www.gsea-msigdb.org) were used in the enrichment analysis. Functional enrichment analysis was performed and visualized using clusterProfiler (4.9.0.2) 4.0[79], and customized scripts. Gene ratio in the enrichment Figures can be defined as k/n where "$k$" is the size of the overlap between your genes of interest and the reference gene set and "$n$" is the number of all unique genes of interest in the reference gene set (universe). Multiple testing was corrected using the Benjamini-Hochberg method and genes and pathways with FDR < 0.05 were considered significant.

## Pseudo-bulk RNA-seq data analysis
A pseudobulk analysis was performed using the AggregateExpression function in Seurat, aggregating gene expression by condition. Differential expression for the pseudobulk dataset was performed using DESeq2 (v1.40.0). Pseudobulk analysis was performed to assess the aggregate impact of LPS treatment in treated versus untreated samples, highlighting the difference in the expression of key LPS-inducible

markers. This allowed us to extract significant differentially expressed markers that are most impacted in the LPS-treated samples at the population level rather than the cellular level, minimizing *p*-value inflation and type-1 error rates.

## Chord plot presentation

The standard GSEA chord diagrams visualize the expression of core genes that comprise significantly enriched pathways in LPS + PA14 samples compared to PA14 samples, as well as highlighting genes that are components of multiple significant pathways. Genes are connected to their associated pathways via colored "chords", with the colors referring to specific pathways as noted in the figure legend. Genes associated with multiple pathways are shown with multiple-colored chords connected to separate pathways. Colored bars just below the gene names represent the fold-change in gene expression in the LPS + 14 samples compared to the PA14 samples, as noted in the figure legend. Genes are ordered based on their logFC, with the genes at the top of the diagram having the highest fold-change in LPS + PA14 versus PA14 samples.

## Quantitative fluorescence intravital lung microscopy (qFILM)

Quantitative fluorescence intravital lung microscopy (qFILM) was performed to visualize phagocytosis of PA-GFP by neutrophils in the lung microcirculation and neutrophil counts in the intact lung of live mice. The qFILM experimental setup and approach have been described in detail previously[80–82]. Briefly, qFILM was performed with a Nikon multi-photon-excitation (MPE) fluorescence microscope (Nikon Instruments Inc; Tokyo, Japan) using an excitation wavelength of 850 nm and an APO LWD 25x water immersion objective with 1.1 NA. Time-series of two-dimensional (2D) qFILM images were collected at ~30 fps using Resonant-scanner. Each field of view (FOV) was 128 μm × 128 μm (~16,384 μm$^2$) with a resolution of ~0.5 μm per pixel in the x-y plane. Fluorescent light received from the sample was collected by different detectors using a series of band pass filters: 450/20 nm (detector 1 for collection of PB), 525/50 nm (detector 2 for collection of PA-GFP), and 576/26 nm (detector 3 for collection of Texas Red Dextran). Prior to thoracic surgery, mice were anesthetized with an intraperitoneal (i.p.) injection of a cocktail containing 100 mg/kg ketamine HCl (Covet-rus, Cat# NDC 11695-0703-1) and 20 mg/kg xylazine (Akorn Pharmaceuticals, Cat# 59399-111-50). Tracheostomy was performed to facilitate mechanical ventilation with ~95% O$_2$ supply and maintenance anesthesia (1.5% isoflurane). The left lung was surgically exposed by serrating three ribs and immobilized against a coverslip using a vacuum enabled micro-machined device as described previously[81]. Just prior to imaging, ~125 μg Texas Red Dextran (MW 70 KDa) (Molecular Probes Inc., Cat# D1830) and 12 μg PB-conjugated anti-Ly6G mAb (Clone: 1A8) (Biolegend, Cat# 127612) were injected into each mouse via the femoral vein for visualization of the pulmonary microcirculation and neutrophils, respectively. qFILM was performed on a mouse for a total period of 60 min and the number of neutrophils in the lung and presence or absence of PA-GFP phagocytosis by neutrophils were assessed for ~3 min in each FOV (total of 12 Fields of View per mouse). Time series of qFILM 2D-images were processed and analyzed using Nikon's NIS-Elements software as described previously[80–82]. First, an image subtraction algorithm was employed to remove autofluorescence and bleed through between channels. Second, signal to noise ratio was improved by using a median filter algorithm followed by smoothing and denoising algorithms. Third, each channel was pseudo-colored as follows: microcirculation as purple, neutrophils as red, and PA-GFP as green to enhance contrast and facilitate visualization. Phagocytosis was defined as colocalization of e-GFP and PB signals. Number of neutrophils was defined as an average number of neutrophils present within the FOV. Effective phagocytosis by neutrophils was measured as average of colocalized e-GFP and PB signals in the Fields of View. Means were compared using unpaired students' *t* test with Welch's correction. Percentages were compared using chi square distribution test.

## Analyses of ETA samples

For all collected samples, we performed a 2-fold dilution with Sputasol (Thermo Fisher Scientific, Cat# SR0233A), followed by centrifugation (375 $g$ × 5 min) and mixing supernatant with PBS to final 20-fold dilution. Pierce bicinchoninic acid assay was used to quantify total protein concentration and a colorimetric assay for urea was used (Abcam, Cat# ab102536). Galectin-3 and neutrophil elastase were assayed by ELISA. All plasma biomarkers for inflammatory phenotypic assignments were measured with the same 10-plex Luminex panel (R&D) as previously described[46]. We analyzed raw ETA biomarker values, as well as normalized values by total protein or urea concentration in each sample to account for any variability in dilution during sample acquisition. Raw ETA biomarker levels were adjusted for protein and urea levels which yielded consistent results. All analyses were performed in R v.4.2.0.

## Statistical analysis and reproducibility

The student's unpaired 2-tailed *t* test with Welch's correction and ordinary one-way ANOVA with the Tukey post-hoc test or Dunnett's test or Sidak's test was used as appropriate. Percentages were compared using the chi-square distribution test. Bacterial load data were log-transformed using log base 10 to adjust for differences in standard deviation prior to analysis. Differences between groups were considered significant at $P < 0.05$. All statistical analyses were performed using GraphPad Prism 9 software (GraphPad Software, La Jolla, CA). For scRNA-seq data, statistical analyses were performed using R version 4.2.1.

## Reporting summary

Further information on research design is available in the Nature Portfolio Reporting Summary linked to this article.

## Data availability

All scRNA-seq generated in this study are available in the GEO with accession number GSE237646. Source data are provided with this paper.

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

## Acknowledgements

We thank all patients who participated in this study and all research coordinators who helped with patient enrollment. We thank Li Fan for preparing the gene expression libraries for the scRNA-seq work, Jie Chen. Kathryn Dalton for technical assistance with the mouse work and Michael Gorry for initial analysis of the scRNA-seq data. This work was supported by NIH grants P01 HL114453 (to P.R., R.K.M., J.S.L., V.E.K. and B.J.M.), K23 HL139987 (to G.D.K.), the R01 grants HL128297, HL141080, HL166345, and an American Heart Association grant 23TPA1074022 (to P.S.), Career Development Award Number IK2 BX004886 from the US Department of Veterans Affairs Biomedical Laboratory R&D Service (to W.B.), and a T32 grant 5T32AI089443 (to Dr. Mark Shlomchik). Computational analysis and storage in this study was supported in part by the University of Pittsburgh Center for Research Computing through the resources supported by the NIH award S10OD028483.

## Author contributions

S.D. designed and performed experiments, analyzed data, and wrote the manuscript. T.W.K. designed and performed intravital imaging of phagocytosis by neutrophils in mouse lungs, analyzed data, and wrote the imaging data. B.T.S. performed all computational analysis of scRNA-seq data, prepared all related Figures, and wrote descriptions of the data. W.B. provided patient samples and interpreted clinical data. S.H. provided excellent technical support in all animal experiments. A.P. participated in animal experiments, S.L.K. provided helpful comments and prepared a schematic of the key findings. K.C. supervised all steps in preparation of the gene expression libraries and downstream sequencing. J.S.L. and R.K.M. provided helpful comments for the study, critically reviewed and revised the manuscript. V.E.K. collaborated on the scRNA-seq work and reviewed the manuscript. D.R. provided oversight on analysis of the scRNA-seq data and critically reviewed all descriptions of the data. B.J.M. interpreted clinical data and critically reviewed and revised the manuscript. P.S. helped design and provide oversight on the intravital imaging experiments and reviewed the manuscript. G.D.K. provided patient samples, analyzed and interpreted clinical data, and wrote the manuscript. A.R. designed experiments, supervised the study, interpreted the data, and wrote the manuscript with input from co-authors. P.R. conceived and supervised the study, designed experiments, interpreted the data, and wrote the manuscript with input from co-authors.

## Competing interests

The authors declare no competing interests.

## Additional information

[1]Division of Pulmonary, Allergy, Critical Care, and Sleep Medicine and Acute Lung Injury Center of Excellence, Department of Medicine, University of Pittsburgh School of Medicine, Pittsburgh, PA 15213, USA. [2]VERSITI Blood Research Institute and Medical College of Wisconsin, Milwaukee, WI 53233, USA. [3]Department of Pediatrics, Division of Health Informatics, University of Pittsburgh School of Medicine, Pittsburgh, PA 15224, USA. [4]Veteran's Affairs Pittsburgh Healthcare System, Pittsburgh, PA 15240, USA. [5]Division of Pulmonary and Critical Care Medicine, Department of Medicine, Washington University School of Medicine, St. Louis, MO 63110, USA. [6]Department of Medicine, The Ohio State University (OSU), Columbus, OH 43210, USA. [7]Department of Environmental and Occupational Health, University of Pittsburgh, Pittsburgh, PA 15261, USA. [8]Department of Immunology, University of Pittsburgh School of Medicine, Pittsburgh, PA 15213, USA. [9]These authors contributed equally: Tomasz W. Kaminski, Brent T. Schlegel, William Bain. [10]These authors jointly supervised this work: Georgios D. Kitsios, Anuradha Ray & Prabir Ray. ✉e-mail: rayp@pitt.edu

