## [Peer Review File · Nature Communications]

REVIEWER COMMENTS

Reviewer #1 (Remarks to the Author):

In their study, Das et al. present noteworthy findings indicating that a low dose of LPS triggers the accumulation of a distinct population of neutrophils. This population displays a unique enrichment in both phagocytosis-associated genes and a set of cell-killing genes, notably *Lgals3*, which encodes the antibacterial protein galectin-3. These findings underscore the potential of galectin-3-expressing neutrophils as a means of defense against lethal infections and respiratory failure. The key observations of the study are as follows:

1. Mice previously exposed to a single low dose of LPS exhibit increased protection against lethal PA14 infections.
2. Single-cell RNA-seq analysis unveils an LPS-induced expansion of a specific neutrophil subset expressing *Lgals3*.
3. Intravital imaging demonstrates that LPS pre-treatment enhances neutrophils' capacity for efficient phagocytosis and cell killing.
4. Clinical investigation reveals higher galectin-3 levels in endotracheal aspirates (ETAs) from survivors compared to non-survivors of acute respiratory failure.
5. Mouse models lacking galectin-3 exhibit elevated lung bacterial burden along with suppressed IFN- γ -expressing neutrophils.

Overall, this study presents intriguing and significant insights. The experimental design appears well-conceived, and the procedures are logically executed. However, while the role of galectin-3 in bolstering host defense against bacteria is compelling, there is a need for more thorough experimental characterization of the galectin-3-expressing neutrophil population (N3). Additionally, the study's focus is on the bacteria-killing capability of the N3 neutrophil population. However, acute respiratory distress syndrome (ARDS) can arise from factors other than bacterial infection, such as trauma and aspiration. While the study illustrates that galectin-3 levels in lower respiratory tract samples predict survival and correlate with markers of neutrophil activation in patients with acute respiratory failure (Fig. 6), the specific relationship of these findings to bacterial infection remains to be clarified, as this information is not provided in the current study.

Other comments:

1. Even at a low dosage, LPS alone elicits significant changes in neutrophil function and gene expression. It is recommended to include an LPS+PBS group as a control in all experiments, allowing for a proper baseline comparison.
2. Figure 2C and 7F. While all neutrophils in the WT group appear to be IFN- γ positive, the authors mention in the discussion that they only detected a small subset of IFN- γ positive neutrophils in PA14-infected mice (page 15, second paragraph). FMO controls are needed to ensure accuracy in gating the true positive population.
3. Figure 4D. The single-cell data indicates a greater presence of *Lgals3* in neutrophils from PA14 mice (N1, N2, and N4, all enriched in *Lgals3*) compared to LPS+PA14 mice (N3 only). However, the authors assert that neutrophils from the LPS-pre-exposed group exhibit superior phagocytosis. To reconcile this

disparity, a straightforward in vitro experiment could be designed to directly compare the phagocytic and bactericidal capabilities of neutrophils from both groups of mice.

4. Within Figure 7A, a marked reduction in bacterial burden is observed in both WT and Galectin-3 KO mice following LPS pre-exposure. Does LPS pre-exposure still enhance bacterial clearance in the absence of galectin-3? This needs to be discussed.

5. The authors propose "Asprv1" as a neutrophil-specific marker. To substantiate this claim, additional citations are needed.

6. Are neutrophils the major source of galectin-3? How does the presence of galectin-3 on or within neutrophils, in comparison to other galectin-3-producing cells, contribute to the promotion of bacteria-killing mechanisms? An in-depth discussion of these aspects could enrich the study's insights.

7. They analyzed endotracheal aspirate from ARF and ARDS patients and found higher galectin 3 levels in survivors compared to non survivors. Interestingly, LRT neutrophil elastase was highly co-related with galectin-3 in survivors. However, several studies showed that neutrophil elastase (NE) can lead to the degradation of the extracellular matrix (ECM), potentially contributing to conditions like chronic obstructive pulmonary disease (COPD) (e.g. Cell. 2019 Jan 10;176(1-2):113-126.e15. doi: 10.1016/j.cell.2018.12.002.)

Reviewer #2 (Remarks to the Author):

This manuscript uses mouse models, extensive scRNA-seq, intravital imaging and some human confirmation studies to study a role of LPS pre-treatment in the progression of *Pseudomonas aeruginosa* lung infection. The work identifies a different neutrophil response in mice pre-treated with LPS 3 days prior to challenge with *P. aeruginosa*, with a focus on the role of galectin-3 as a mediator of apparent LPS-mediated protection.

The work is an impressive collection of techniques, analysis and approaches. The concept is interesting and deserving of interest. As a microbiologist I do however have concerns as to some of the justifications for the experimental approaches.

Major issues:

1. The sc-RNA seq work and the subsequent analysis is only performed on the *P. aeruginosa* challenged mice (4 hours post infection) and *P. aeruginosa* challenged mice which had been pre-exposed 72 h to LPS. To infer any role of LPS in modulating/shaping the immune response, the cohort exposed only to LPS needs to be analyzed and compared. As it stands the two groups represent two very different stages of infection, one which has been infected for 4 h (PA) and one group which has had ongoing inflammation for 72 h + 4 h (LPS + PA) – it is unsurprising there are dramatic differences in the immune cells in the two groups. The differences seen in inflammatory signaling in Fig 1 highlights this, with the LPS alone cohorts showing different inflammatory kinetics even between 24 and 72h, without *P. aeruginosa* challenge.

2. The concept that LPS could be protective challenges many years of research into the role of LPS in lung damage, allergy, shock etc. LPS induces lung inflammation and the recruitment of cells, typically via a

TLR4 mediated pathway. In this work the authors are studying the response to infection in a naïve vs already inflamed lung, are the findings in any way specific to LPS induced inflammation? Could any of the findings be linked to specific LPS mediated pathways? Secondary to this, can the authors justify their choice of LPS, why use an E. coli LPS from a gastrointestinal bacterial source rather than LPS from P. aeruginosa? (Pandur E, et al Int J Mol Sci. 2021)

3. The n numbers and biological replicates for the groups in particular figure 1 are very low and inconsistent. Figure 5 seems to be 2 mice, Figure 7 from a single experiment, is this correct? Were power-calculations made prior to experiments?

Minor issues:

1. Choice of bacterial strain, what is the source of the chosen strain? PA-14 is typically considered 'hypervirulent' compared to PA01. Is PA-GFP used in the imaging work a PA-14 strain or what is it's background? Strain nomenclature and sources should be added.
2. The time-frames of the infections in Figure 1 are difficult to understand in the results section and needs clarification. Were the mice infected with P. aeruginosa for 4 or 6 hours, this differs from the results to the figure legends? Were the non-LPS challenged animals obviously sick by 8h or 4h?
3. In figure 5, can the increased co-localization of PB and PA-GFP not be explained by the increased number of PMNs in the pre-inflamed LPS treated group? The supplementary video related to this figure did not work.
4. From which mice were the BAL cells collected in figure 5f-h? A new group or?
5. Galactin-3 KO mice have previously been shown to be more susceptible to lung infection (aspergillus) due to deficits in neutrophil mobility and extravasation (Snarr et al PloS Path 2020). How does the data here correlate to previous work?
6. The loss of INF γ in the KO mice neutrophils is indeed interesting and could have been followed up – is this LPS specific?

Reviewer #3 (Remarks to the Author):

In this paper, Das et al. study the urgent need for understanding mechanisms that enhance the host immune response to pathogens in the era of rising rates of drug-resistant pathogens.

The authors provide a mechanistic insight into a prior, well-documented, observations that intratracheal pre-exposure of mice to a low dose of LPS protects mice from lethal pulmonary infection by P. aeruginosa (PA). Using single-cell RNA-seq, the authors identified macrophage and neutrophil subsets that are enriched in LPS-pretreated mice. The authors further focus on N3 neutrophils population, which

expresses phagocytosis and cell-killing related genes. One of the genes that appeared from their analysis was *Lgals3*, which encodes the multifunctional antibacterial protein galectin-3. Phagocytosis and cell killing by neutrophils, as well as galectin-3 protein, were experimentally validated. Using human samples, the authors show higher levels of galectin-3 in survivors vs. non-survivors, as well as a correlation between galectin-3 levels and neutrophil marker. The work here shows the important role of neutrophils and galectin-3 in LPS-mediated protection from PA infection.

This manuscript is timely, dealing with a very important subject, is well-written, and uses multiple complementary tools. The data is convincing, and the interpretation is in alignment with the results with no over-estimation.

I would ask the authors to address the following points:

Major points:

1. The authors are missing an important integration of what is known in the field and the data presented here. Two main topics are: (1) LPS is known to cause immune tolerance. Do the authors consider their finding as tolerance? (2) How do the findings here correspond to the growing knowledge of trained immunity? The authors should introduce these topics, as well discuss their result in this context:
 - a. As the animals are septic 4h after the infection, the LPS protection can be a reduced pro-inflammatory response, unrelated to the specific bug. It would be interesting to compare their results to lower CFU for infection and longer survival time of PA infection – is there still a protection? This result can indicate some more specific protective effect. In any case – these results could be an interesting addition to their findings.
 - b. Their findings of increased Stat1 and IFN γ (Fig. 1f) seems to support a trained immunity response.
 - c. Conversely, the IL-10 behavior is different from classic LPS- tolerance.
 - d. Similarly – how their results of Type-I IFN relate to known effects of trained immunity.
2. The technical details of the Single-cell experiment are unclear:
 - a. - the main text doesn't mention the biological replicates. how similar are the replicates? Are N3 and IM2 proportions similar between the biological replicates, or dominated by one replicate? - The authors need to perform an analysis on the different replicates in the main figure
 - b. Even in the methods, it is not clear if hushing was used or different 10X lanes. Also.
 - c. Some of the figures format are difficult to follow. It is not clear what are the populations that are indicated in supp Fig. S2a. It is challenging to associate between the 18 populations the authors got in their analysis and the cluster annotation that they gave after joining few populations (supp 3). Naming the 18 clusters could help. In supp. Fig. S2b – please make a dotplot only of relevant genes. Fig. 4e is very hard to follow, and would ask to simplify their visualization. It takes a lot of effort to understand the logic.
3. Fig. 6- human samples: it is not clear how many of the patients have PA infection? Are the authors claiming that Galectin-3 is associated with positive survival rate for any acute respiratory distress syndrome?

Minor:

1. Supp 1: explain what are the gated populations.
2. Explain how the cell counts were performed using a flow cytometer- did the authors use counting beads?
3. FACS plots are in low resolution.
4. On pg. 5, the authors claim:
'Individually, both LPS and PA14 induced significant increases in the numbers of CD11b+Ly6G+ neutrophils and CD11b+CD64+CD24- interstitial macrophages (IMs), both cell types having high phagocytic functions, with neutrophil numbers being only slightly higher in the LPS+PA14 mice (Fig. 2a, b).'
- The figures don't show an increase of CD11b+CD64+CD24- interstitial macrophages (IMs) in PA14.
5. On pg. 7 the authors write:
'As depicted in the volcano plots, some genes were upregulated in multiple clusters which included...'
Should be violine plots and not volcano plots.
6. It is difficult to see colorbar changes in figures (e. g. 3b, 4c) due to the scale of colors. Please make a more precise scales and make sure of color consistency (high and low) along the figures.
7. Fig. 5e- is presumably from scRNAseq data; however, this is not mentioned in the test/fig. legend
8. in supp table 3, 30-day mortality, n (%) for At-risk for ARDS appears as 10 (18.9), which is not in agreement with the number of dots in fig. 6
9. In the discussion, the authors wrote:
'We do not know whether the major source of LPS-induced Ifng was the N3 population.' (page 15). This is followed by a paragraph that is not clear to me. Why the authors can't check IFNg expression in their single-cell data to indicate the relevant population?
10. In the discussion, please comment on how long the LPS effect lasts (the authors show 3-days, how about 1 week? 1 month?)
11. Fig. 8 legend (the model) is very poor, as well as its description in the main text. The authors need to elaborate on their figure. E.g, what does the question mark stand for In the alveolus populations? It would help to indicate only the important findings in this illustration.

Reviewer #4 (Remarks to the Author):

The work from Das, Ray, and colleagues uses multiple approaches to examine the protective effects of LPS during *P. aeruginosa* infection (PA14). Using ScRNASeq of lung tissue they describe the immune populations of Cd11b+ cells during infection with PA14 in naïve or LPS-conditioned mice and identify discrete cell populations. Based on the single-cell data the authors claim that Galectin-3 mediated neutrophil host defense functions critical for improved survival. The authors include intravital microscopy demonstrating a protective effect of LPS pre-conditioning and patient-level data showing associations between Galectin-3 levels and survival in patients with acute respiratory failure. Finally, they subject Galectin-3 KO mice to infection and demonstrate increased bacterial burden in the absence of Galectin 3. While the authors should be commended for the numerous approaches taken to validate the

hypothesis, this manuscript is currently lacking cohesiveness and solid evidence demonstrating a role for Galectin-3 and the newly identified neutrophil subset in mediating the protective effects of LPS during PA14 infection. Moreover, while the authors should be commended for including patient-level data, but the claim that Galectin-3 can serve as a prognostic biomarker in ARF is unsubstantiated.

General comments: Several concerns raised by the manuscript preclude acceptance of the manuscript in its current form.

1. While the authors provide an extensive description of the ScRNASeq data, the claim that the protective effects are mediated by N3 population is not substantiated, the authors failed to examine the IM2 population which is also induced in the pre-conditioned mice and carries a robust cell killing and phagocytic signature.

2. While the importance of neutrophil-mediated killing and bacterial clearance is not denied, there are no specific studies to link the Galectin-3 findings with neutrophils. Ideally, deletion of Galectin 3 in neutrophils or macrophages would be needed to better delineate how Galectin-3 deficiency impairs host defense in the LPS-PA14 mice.

3. The title of the paper claims Galectin-3 mediates protection from lethal bacterial infection in mice. Unless I am missing this data, I do not see evidence of improved survival in the mouse models in the absence of Galectin-3.

Comments:

1. Innate immune training has been classically described in monocytes and tissue macrophages. Have the authors considered examining the IM2 population which is also robustly induced by the LPS pre-conditioning. The specific focus on neutrophils while ignoring the IM2 population limits the strength of this manuscript.

2. The authors provide correlative data in the patient cohort and show elevated cell associated Galectin-3 in the LPS+PA14 mice. Can the authors do additional studies to help determine 1. The cellular source of the Galectin 3, 2. If the cell-associated Galectin-3 is secreted and adherent to the membrane, v intracellular v in the BAL (non-cellular fraction) in the mice. This would be important as Galectin-3 is thought to function as a DAMP and can be derived from numerous cell types. Studies to better delineate the specifics of Galectin-3 localization in the LPS-pre-conditioned model would help elucidate the mechanism of protection offered by Galectin-3.

3. Did exogenous Galectin-3 administration rescue the knockout mice?

4. Throughout the authors describe the BAL cell numbers and bacterial burden but with the exception of figure 1 showing differences in permeability and cytokine responses there are few other characterizations on the protective effect of LPS on PA14 infection. Can the authors provide some histology (if available).

5. Many of the studies were only done 2 times with few numbers of mice (3-4), Figure 1, additionally, Figure 7 demonstrating the effects of Galectin-3 deletion is only 1 study with 4 mice. Please provide adequate numbers of mice and experimental replicates to ensure robustness of the data.

6. Using qFILM, the data demonstrates that LPS training increases phagocytosis by PMN. Would it be possible to examine the Galectin-3 KO mice using the same methodologies? Alternatively can the authors conduct ex-vivo studies demonstrating impaired phagocytosis of bacteria in the absence of Galectin-3 on neutrophils or macrophages.

7. There is an expansive literature on Galectin-3 and IFN γ regulation in tumor biology. How do the authors reconcile their findings of decreased IFN γ secreting PMN in the absence of Galectin-3 with the large body of evidence showing a role of Galectin-3 in dampening IFN? One would imagine there would be more IFN γ present in the absence of Galectin-3. While the authors show decreased intracellular IFN γ in the Cd11b/Ly6G/CD14 pmn it'd be interesting to see if systemic or lung levels of IFN γ are altered in the absence of Galectin-3.

8. The patient level data demonstrating the correlation between neutrophil elastase and galectin-3 may be misleading as this does not prove that the LRT Galectin 3 is derived from the neutrophils. Would try to rephrase the conclusions based on the correlative data. Would certainly remove the phrase "our data suggest a prognostic value of Galectin-3". Prognostic of what, without additional data it'd be hard to claim that the ETA Galectin-3 is a prognostic biomarker in ARDS.

9. There are several references made to post-viral COVID ARDS (in terms of hyperinflammation/IL-6 in the discussion). Would ask that the authors instead consider placing their findings in context of the large body of evidence examining the "hyperinflammatory" ARDS endotypes.

We thank the reviewers for the kind and positive comments, and very much appreciate their constructive criticisms. We have addressed the questions raised to the best of our ability with additional data included from new experiments performed.

The following is our point-by-point response to the comments:

Reviewer #1

We thank the reviewer for the many positive comments and enthusiasm for our study. Our response to the specific questions raised is as follows:

1. Even at a low dosage, LPS alone elicits significant changes in neutrophil function and gene expression. It is recommended to include an LPS+PBS group as a control in all experiments, allowing for a proper baseline comparison.

Response. Ultrapure LPS was dissolved in LPS-free ultrapure water to prepare a stock solution and diluted further in LPS-free ultrapure PBS for working stocks. Naïve mice received the same PBS.

2. Figure 2C and 7F. While all neutrophils in the WT group appear to be IFN- γ positive, the authors mention in the discussion that they only detected a small subset of IFN- γ positive neutrophils in PA14-infected mice (page 15, second paragraph). FMO controls are needed to ensure accuracy in gating the true positive population.

Response. Please see inclusion of FMO controls in Figs. 2c and 7f.

3. Figure 4D. The single-cell data indicates a greater presence of *Lgals3* in neutrophils from PA14 mice (N1, N2, and N4, all enriched in *Lgals3*) compared to LPS+PA14 mice (N3 only). However, the authors assert that neutrophils from the LPS-pre-exposed group exhibit superior phagocytosis. To reconcile this disparity, a straightforward in vitro experiment could be designed to directly compare the phagocytic and bactericidal capabilities of neutrophils from both groups of mice.

Response. The revised color scale in all Figs related to scRNA-seq data has helped to show that while *Lgals3* is expressed in the PA14 mice in N1, N2 and N4 clusters, the level of its expression in the clusters N3, IM2 and IM3, which all show enrichment of the cell killing pathway in LPS+PA14 mice vs PA14 mice, is appreciably higher in the former group (Fig. 4d) with cell counts for N3 and IM2 being much higher in this group too (Fig. 3a). Also, as discussed in our manuscript, while *Lgals3* is expressed variably in all neutrophil populations in both groups, expression of *Gbp2*, which was previously shown to partner with galectin-3 for bacterial killing in phagosomes (Feeley et al., PNAS, 2017), is selectively upregulated in the N3 population among the four neutrophil populations. It is challenging to perform in vitro phagocytosis experiments with primary neutrophils which may also not be representative of in vivo events when taken out of context. However, the bacterial counts in BAL cells comprising mostly neutrophils, being >95% in PA14 mice and >98% in LPS+PA14 mice, show 2 log-fold lower counts in cells from LPS-treated mice (Fig. 5m) showing differential bacterial clearance capabilities in the two groups of mice.

4. Within Figure 7A, a marked reduction in bacterial burden is observed in both WT and Galectin-3 KO mice following LPS pre-exposure. Does LPS pre-exposure still enhance bacterial clearance in the absence of galectin-3? This needs to be discussed.

Response. As shown in Fig. 7a, bacterial load in the lung tissue at 4h post-infection is significantly higher in galectin-3 KO LPS+PA14 mice (1.5 log-fold) compared to that in WT LPS+PA14 mice but is lower than that observed in WT PA14 mice (without LPS pre-exposure). Independent experiments have yielded similar data and pooled results are depicted. In addition, we present *new data* (Fig. 7h) showing higher bacterial burden in the BAL cells (majority being neutrophils) isolated from the galectin-3 KO mice compared to those isolated from WT mice (both LPS pre-exposed). This shows that absence of galectin-3 impairs bacterial clearance by neutrophils and likely also by IMs in the lung tissue. However, it is important to note that in contrast to their LPS-exposed WT counterparts, while LPS-exposed galectin-3 KO mice reproducibly appear very sick initially (4-6 h after infection), they slowly recover 24 h after infection. Collectively, our data show that while galectin-3 is important for early bacterial clearance (likely in partnership with Gbp2), other anti-bacterial molecules that comprise the cell killing pathway such as Cxcl9, Cxcl10 and C3 in N3 may help in bacterial clearance at later time points. This early anti-bacterial function of galectin-3, however, may provide the survival advantage that we observe in critically ill ICU patients with higher galectin-3 levels in their endotracheal aspirates (ETAs). Alternatively, compared to other anti-bacterial proteins, galectin-3 may play a more crucial role in host defense in critically ill humans.

5. The authors propose "Asprv1" as a neutrophil-specific marker. To substantiate this claim, additional citations are needed.

Response. It was originally identified as a neutrophil-expressed gene in 2017 and published in JCI Insight which we have cited. These authors were granted an international patent and a US patent in 2019 (WO2019014744 and US20200355684 respectively). Additional references are by Skinner et al., (PMID: 36248905) and Nederlof et al (PMID: 35911749) which are included in the revised manuscript.

6. Are neutrophils the major source of galectin-3? How does the presence of galectin-3 on or within neutrophils, in comparison to other galectin-3-producing cells, contribute to the promotion of bacteria-killing mechanisms? An in-depth discussion of these aspects could enrich the study's insights.

Response. Please see response to question 3. Our data overall suggest that galectin-3 has an important role in bacterial killing in neutrophils, as also suggested by GSEA of scRNA-seq data. In the N3 population, galectin-3-Gbp2 interaction may have an important role in bacterial eradication which may also be true in the IM populations IM2 and IM3 that show increased Gbp2 expression. However, it is important to note that in the airspaces, neutrophils are critical for bacterial clearance since IMs are typically not present in these compartments. In BAL of LPS+PA14 mice, neutrophils were ~100X more abundant than in PA14 mice (Fig. 5i) and this is also reflected in the difference in bacterial load in the BAL cells/neutrophils in the two groups (Fig. 5m). These are discussed in the revised manuscript.

7. They analyzed endotracheal aspirate from ARF and ARDS patients and found higher galectin 3 levels in survivors compared to non survivors. Interestingly, LRT neutrophil elastase was highly co-related with galectin-3 in survivors. However, several studies showed that neutrophil elastase (NE) can lead to the degradation of the extracellular

matrix (ECM), potentially contributing to conditions like chronic obstructive pulmonary disease (COPD) (e.g. Cell. 2019 Jan 10;176(1-2):113-126.e15. doi: 10.1016/j.cell.2018.12.002.)

Response. We assayed NE as a surrogate for neutrophils. We agree with the reviewer that excess NE can cause lung injury. However, the positive correlation of NE with survival of critically ill patients highlights the beneficial role of neutrophils in host defense. An early influx of neutrophils at the infected site is essential in host defense, with some populations, like N3, specifically programmed to do so as evident in our study. However, they can cause lung injury unless subsequently cleared to allow resolution of inflammation and restoration of homeostasis. Indeed, we previously showed a role for myeloid regulatory cells in neutrophil efferocytosis and resolution of inflammation, an impairment of which induced lung injury (Chakraborty et al., Nature Comm., 2017; PMID: 28074841).

Reviewer #2

We thank the reviewer for expressing interest in our study. Our response to the specific questions is as follows:

Major issues:

1. The sc-RNA seq work and the subsequent analysis is only performed on the *P. aeruginosa* challenged mice (4 hours post infection) and *P. aeruginosa* challenged mice which had been pre-exposed 72 h to LPS. To infer any role of LPS in modulating/shaping the immune response, the cohort exposed only to LPS needs to be analyzed and compared. As it stands the two groups represent two very different stages of infection, one which has been infected for 4 h (PA) and one group which has had ongoing inflammation for 72 h + 4 h (LPS + PA) – it is unsurprising there are dramatic differences in the immune cells in the two groups. The differences seen in inflammatory signaling in Fig 1 highlights this, with the LPS alone cohorts showing different inflammatory kinetics even between 24 and 72h, without *P. aeruginosa* challenge.

Response. Our experiments were designed to address how just one time pre-exposure to a low dose of 20 µg of LPS completely protects mice from a lethal infection by PA14. Since the bacterial load progressively decreased from day 1 to day 3 after LPS exposure, it suggested a dynamic mechanism that helped to prepare the lung for a subsequent infection that in fact made a difference between life and death in the mice. Therefore, using the approach of scRNA-seq, we asked whether LPS pre-exposure promoted one or more cell populations and whether they expressed gene sets associated with phagocytosis and cell killing. Since the lung bacterial load after infection was lower on day 3 as compared to that on day 1 from the time of the initial LPS exposure (Fig. 1b), we focused on day 3 post exposure for this work. As the reviewer has noted, we did compare the host response to LPS alone vs LPS+PA14 by way of expression of cytokine genes and effector responses (ISGs, STAT1) and noted overall similarities in the profiles other than a small boost in the expression of some genes in the LPS+PA14 mice compared to that in LPS mice (example-*Ifng* although STAT1 expression was similar). With these results, we proceeded to compare the lung immune cells in mice that failed to recover (PA14 alone) vs those that showed complete protection (LPS+PA14) since that was the key question we wished to answer. We would like to point out that we actually did not see dramatic differences in the immune cells in the two groups except for two cell clusters, N3 and IM2, that selectively expanded under LPS exposure (Fig. 3). Most importantly, the scRNA-seq data showed that the time period of 4 h of PA14 infection, which was common between the LPS+PA14 and PA14 mice when mice were taken down for

analysis, was insufficient to cause accumulation of N3 and IM2 although was sufficient for the accumulation of other cells in very similar numbers. Thus, LPS pre-exposure clearly accounted for the higher abundance of N3 and IM2. Again, the lower lung bacterial load on day 3 vs day 1 after LPS exposure suggested an LPS-mediated increase over time in N3 and IM2 although this needs to be examined in future studies.

2. The concept that LPS could be protective challenges many years of research into the role of LPS in lung damage, allergy, shock etc. LPS induces lung inflammation and the recruitment of cells, typically via a TLR4 mediated pathway. In this work the authors are studying the response to infection in a naïve vs already inflamed lung, are the findings in any way specific to LPS induced inflammation? Could any of the findings be linked to specific LPS mediated pathways? Secondary to this, can the authors justify their choice of LPS, why use an *E. coli* LPS from a gastrointestinal bacterial source rather than LPS from *P. aeruginosa*? (Pandur E, et al Int J Mol Sci. 2021)

Response. Yes, we completely agree that LPS can cause injury at high doses and there are many studies that have studied LPS-induced injury of the lung and other organs. However, as we have already cited, the protective effect of a low dose of LPS to subsequent infection was described decades ago and confirmed in subsequent studies by other investigators, as also by us in the present study. We have shown the ability of LPS to induce *Aoah* (Fig. 1), which plays an important role in preventing the toxic, injurious effects of LPS but allows the host to respond to subsequent infections. We have cited additional studies that have shown the importance of *Aoah* induction by LPS in limiting lung injury (Zou et al., eLife, 2021). The precise downstream mechanisms that promote this protective response via *Aoah* induction is not well understood at the present time. In the next phase of our work, we would investigate whether in our experimental system *Aoah* is required for protection from PA14 infection. With regard to the choice of *E. coli* LPS, our question was not so much as to whether the LPS in PA14 *per se* can confer protection but rather can available ultrapure LPS protect mice from death with significance in protection from ventilator-associated pneumonia (VAP), which is most commonly caused by infection by PA.

3. The n numbers and biological replicates for the groups in particular figure 1 are very low and inconsistent. Figure 5 seems to be 2 mice, Figure 7 from a single experiment, is this correct? Were power-calculations made prior to experiments?

Response. All experiments in the revised experiment have been repeated. Specific experiments shown in Figs 1, 5 and 7 have been repeated and the revised Figs reflect that. Original Fig. 5 was derived from 2 independent experiments with 2 mice per group in each experiment. These experiments are very labor intensive and only few mice can be analyzed on a given day. However, we have repeated the experiment with similar results. The data shown in Fig. 7 was from 1 experiment initially with 4 mice in each group since additional galectin-3 KO mice were not available from The Jackson Laboratory at that time. This experiment has been repeated and the pooled data are shown in revised Fig. 7. Yes, power analysis was performed based on the differential load between the LPS+PA14 and PA14 mice which yielded a sample size of 3 in each group.

Minor issues:

1. Choice of bacterial strain, what is the source of the chosen strain? PA-14 is typically considered ‘hypervirulent’ compared to PA01. Is PA-GFP used in the imaging work a PA-14 strain or what is it’s background? Strain nomenclature and sources should be

added.

Response. PA14 was donated as a gift to Dr. Janet Lee by Dr. Zhenyu Cheng, a former postdoc of Dr. Frederick Ausubel at Harvard University before his retirement. PA-GFP was purchased from ATCC and has a backbone of PA01. As we show in Supplementary Fig. 5, the mice overall responded similarly to the two strains with a slightly higher bacterial load in the lungs in response to PA14. The sources of the strains are now included in the revised manuscript.

2. The time-frames of the infections in Figure 1 are difficult to understand in the results section and needs clarification. Were the mice infected with *P. aeruginosa* for 4 or 6 hours, this differs from the results to the figure legends? Were the non-LPS challenged animals obviously sick by 8h or 4h?

Response. In our pilot and subsequent experiments, we consistently observed that mice not pre-exposed to LPS and infected with PA14 were huddled appearing visibly ill by 4-6 h after infection. At later time points of 16-18 h after infection, the PA14 mice appeared near death meeting criteria for euthanasia per our animal protocol (IACUC) rules. Since we did not want any mice to die after infection before we had the opportunity to harvest the lungs and other samples, we always started our tissue harvest promptly at 4 h after infection to complete all processing within the next 2 h.

3. In figure 5, can the increased co-localization of PB and PA-GFP not be explained by the increased number of PMNs in the pre-inflamed LPS treated group? The supplementary video related to this figure did not work.

Response. We apologize that the video did not work. Please use VLC player (works for both Mac and Windows) or Media Player (for Windows only). Fig. 5d shows average colocalization of PA-eGFP and PB per FOV. In Fig. 5e, we have also presented percentage of e-GFP+ neutrophils which considers all neutrophils in the 12 Fields of View.

4. From which mice were the BAL cells collected in figure 5f-h? A new group or?

Response. Mice used for intravital imaging cannot be used for other end-points. Data shown in other panels in Fig. 5 were from different mice.

5. Galactin-3 KO mice have previously been shown to be more susceptible to lung infection (*aspergillus*) due to deficits in neutrophil mobility and extravasation (Snarr et al PloS Path 2020). How does the data here correlate to previous work?

Response. That galectin-3 KO mice are more susceptible to fungal infection aligns with our finding of a protective role of the molecule in bacterial infection. Yes, galectin-3 is a neutrophil chemoattractant as we discussed in our manuscript. However, while loss of galectin-3 made a significant difference in bacterial load in BAL cells of LPS-exposed mice, >98% of them being neutrophils, its loss did not reduce neutrophil numbers in the lung tissue or BAL. In future work, it will be interesting to determine which chemokine(s) mediate neutrophil chemotaxis to the lung and the airspaces in our model.

6. The loss of INF γ in the KO mice neutrophils is indeed interesting and could have been followed up – is this LPS specific?

Response. We completely agree. We would like to determine in future studies whether pre-exposure to ligands of other TLRs, such as TLR2, can also induce a similar response

in neutrophils.

Reviewer #3

We thank the reviewer for the many positive comments and enthusiasm for our study. Our response to the specific questions raised is as follows:

Major points:

1. The authors are missing an important integration of what is known in the field and the data presented here. Two main topics are: (1) LPS is known to cause immune tolerance. Do the authors consider their finding as tolerance? (2) How do the findings here correspond to the growing knowledge of trained immunity? The authors should introduce these topics, as well discuss their result in this context:

Response. This comment is elemental to the findings reported in this manuscript and we agree we should have discussed the data in the context of trained immunity. We thank the reviewer for urging us to discuss our findings in the context of this topic. That repeated LPS exposure leads to immune tolerance was described as early as 1947 by Beeson (cited). This is an important mechanism of immune regulation to prevent tissue damage from an overexuberant inflammatory response. In our work, a single low dose of LPS induced resistance to infection. Given that we found that this one-time LPS exposure helped in bacterial clearance and survival of the mice, we assessed the expression of the enzyme Acyloxyacyl hydrolase (*Aoah*) that specifically facilitates the removal of two fatty acyl chains from the lipid A moiety of LPS and prevents tolerance induction. Drs. Robert Munford and Mingfang Lu pioneered this work (we cited some of their papers) and quoting from one of their publications in JEM (PMID: 30021797)- “importantly, AOA promotes recovery from endotoxin tolerance, a transient refractory state or cellular reprogramming that can follow LPS exposure.” As we mentioned, *Aoah* prevents the toxic effects of LPS, prevents immune tolerance, and allows response to a subsequent infection. We have expanded on this discussion categorically discussing the issue of tolerance. Indeed, our findings relate to trained immunity of neutrophils (and monocytes that differentiate to IMs). The literature has been largely dominated by discussions of training of monocytes. The revised manuscript includes a discussion of this topic.

a. As the animals are septic 4h after the infection, the LPS protection can be a reduced pro-inflammatory response, unrelated to the specific bug. It would be interesting to compare their results to lower CFU for infection and longer survival time of PA infection – is there still a protection? This result can indicate some more specific protective effect. In any case – these results could be an interesting addition to their findings.

Response. While certainly a balanced inflammatory response would be host-protective, we show that the LPS exposure led to efficient bacterial clearance and survival of the mice. Thus, an active mechanism was clearly at play for bacterial clearance. Insights from GSEA in the scRNA-seq data led us to focus on galectin-3 and using the galectin-3 KO mice showing impaired bacterial clearance we demonstrate causality. However, we have acknowledged that other molecules such as CXCL9/10 and complement C3 that are also upregulated in the N3 population and are components of the cell killing gene set may also contribute to defense against a lethal infection by PA14 although galectin-3 appears to be important for reduction in bacterial load early after infection. We agree that studying the impact of a non-lethal dose will be interesting.

b. Their findings of increased Stat1 and IFN γ (Fig. 1f) seems to support a trained immunity response.

- c. Conversely, the IL-10 behavior is different from classic LPS- tolerance.
 d. Similarly – how their results of Type-I IFN relate to known effects of trained immunity.

Response. As the reviewer has commented, the reduced IL-10 response also shows that the animals were not tolerized by LPS. Please note that in Fig. 4a in our pseudobulk analysis of scRNA-seq data, we show *AW12010* to be one of the most differentially expressed genes expressed at a higher level in LPS+PA14 compared to PA14 mice. As discussed, this molecule was shown to suppress IL-10 production but promote host response to infection by inducing IL-12 production that would in turn increase IFN- γ production (Jackson et al., Nature, 2018). It is possible that expression of *AW12010* and *Aoah* are functionally linked which we plan to pursue in the next phase of our work. This is also further discussed in the revised manuscript.

2. The technical details of the Single-cell experiment are unclear:

- a. - the main text doesn't mention the biological replicates. how similar are the replicates? Are N3 and IM2 proportions similar between the biological replicates, or dominated by one replicate? - The authors need to perform an analysis on the different replicates in the main figure

Response. In both the legend to Fig. 3 and in the Methods section we did mention “Three biological replicates each of LPS+PA14 and PA14 samples were subjected to

sequencing.” The three biological replicates from the same condition (i.e. PA14 vs LPS+PA14) were merged using the Seurat function merge and is a standard first step in the single cell analysis pipeline. The merged Seurat object from each condition was in the form of a combined data matrix but we could still tell which replicate the cells come from. UMAP and tSNE plots are standard representations to depict how well the replicates overlap with each other. The split ‘dimplot’ is shown below to represent how consistent the UMAP is when split based on replicates.

The next step in the workflow was the integration of single cell datasets which is an important step. This step matches shared cell types across the conditions (LPS+PA14 and PA14) and boosts statistical power. Most importantly, integration facilitates comparative analysis across the datasets. We used the ‘anchor-based’ integration workflow in our analysis. This is the UMAP figure that we have shown as the main figure as we use it for differential expression analysis, and functional enrichment.

- b. Even in the methods, it is not clear if hushing was used or different 10X lanes.

Response. In the methods section, we mentioned that replicates were processed for sequencing individually. This study did not use the cell hashing-based sequencing protocol.

c. Some of the figures format are difficult to follow. It is not clear what are the populations that are indicated in supp Fig. S2a. It is challenging to associate between the 18 populations the authors got in their analysis and the cluster annotation that they gave after joining few populations (supp 3). Naming the 18 clusters could help. In supp. Fig. S2b – please make a dotplot only of relevant genes. Fig. 4e is very hard to follow, and would ask to simplify their visualization. It takes a lot of effort to understand the logic.

Response. We have stated in the text “We identified 14 discrete cell types based on distinct markers and after merging similar clusters noted for IM2 (4 and 5), N3 (7 and 8) and N4 (0 and 1) (Fig. 3a, b, and S2a, b).” We are not sure what else the reviewer would like to see. All of the genes shown in Fig. S2b are the top positive markers for each of the initial clusters generated using FindAllMarkers from Seurat. We opted to include this to contextualize the automated and manual annotations, as well as to show the similarity in top marker expression for merged clusters. Please refer to the subset dot plot (Fig. 3b) for the simplified figure highlighting the expression of key markers that were most informative for our annotations. Fig. 4e depicts standard Chord plots. Please see our description in the methods section of the MS text explaining how the chord plots should be read.

3. Fig. 6- human samples: it is not clear how many of the patients have PA infection? Are the authors claiming that Galectin-3 is associated with positive survival rate for any acute respiratory distress syndrome?

Response. Thank you for this opportunity to provide further details in our dataset. Our human cohort is inclusive of all etiologies of ARDS and Acute Respiratory Failure, and we perform detailed consensus clinical phenotyping among physician-scientist experts for the etiology and diagnosis of ARF in each case.

To address the question of the causal microbiology in patients with ARDS or ARF from direct lung injury, we reviewed all relevant clinical microbiology obtained within 48 h of research sample acquisition. We have provided the results of this analysis in a detailed Supplemental figure. We found that galectin-3 levels were not associated with individual organisms (classified in major pathogen or commensal categories) but were significantly higher when Gram stains detected the presence of neutrophils in the specimen.

Importantly, this observation also aligns well with prior studies (cited in the manuscript) that show high neutrophil numbers in bronchoalveolar lavage (BAL) associating with bacterial infection in patients.

We added the following section in Results:

There was no significant difference in LRT galectin-3 levels between patients with ARDS (median = 146.7, interquartile range [83.4-613.1] ng/ml) vs. at-risk for ARDS (119.1 [71.3-193.1], $p=0.25$), as well as no differences between patients with direct vs. indirect risk factors for lung injury. Among 65 patients with direct lung injury risk factors (pneumonia: 56; macro-aspiration: 7; inhalational injury: 4; pulmonary vasculitis: 1), we found no systematic difference of LRT galectin-3 levels when patients were stratified by the organisms isolated in clinical microbiologic cultures of LRT biospecimens (ETA or BAL) obtained as part of the diagnostic workup by the treating clinicians (Supplementary Fig. 6a). However, presence of neutrophils in Gram Stain examination of these clinical LRT biospecimens was strongly associated with higher galectin-3 levels in ETA research biospecimens (Supplementary Fig. 6b). Thus, LRT galectin-3 levels in our research biospecimens related to neutrophilia and not specific organisms in clinical LRT biospecimens.

We provide further details in the Supplement and the figure legend for this analysis:

Supplementary Figure 6. Clinical microbiologic culture results of lower respiratory tract specimens and associations with galectin-3 levels in research endotracheal aspirate samples. We examined all available microbiologic cultures from lower respiratory tract (LRT) biospecimens, including endotracheal aspirates (ETA) and bronchoalveolar lavage (BAL) samples, as directed by the treating clinicians, when obtained within 48 h of a research ETA biospecimen in which we measured galectin-3 levels. From clinical microbiology laboratory results, we recorded information on neutrophil presence on Gram stains, as well as the isolated and reported organisms, classified as Gram-positive pathogens, Gram-negative pathogens, polymicrobial infections (involving both Gram-positive and Gram-negative pathogens), normal respiratory flora (NRF), yeast (*Candida* species) or no microbial growth. Results are presented in patients with direct lung injury risk factors (n=65), for whom clinical microbiology of LRT biospecimens was important for infection diagnosis and antimicrobial treatment guidance. **a** No significant differences of LRT galectin-3 levels by isolated organisms in clinical microbiology analyses. Gram-positive pathogens included *S.aureus* (n=8) and *Streptococcus spp* (n=2), whereas gram-negative pathogens included *P.aeruginosa* (n=1) and *Enterobacteriaceae* (n=4). **b** Presence of neutrophils on gram stain analysis of clinical LRT biospecimens was strongly associated with galectin-3 levels in synchronous (within 48 h) research ETA biospecimens.

With regard to galectin-3 levels and patient survival, yes, we have found an association between ETA galectin-3 levels and survival of patients with ARF. As stated in Results “Among all patients followed for 30-day survival, survivors had significantly higher LRT galectin-3 levels compared to non-survivors (Fig. 6a), and this difference was driven mostly by patients with ARDS (Fig. 6b).”

Minor:

1. Supp 1: explain what are the gated populations.

Response. Although the text includes the names of all gated populations, we have included the names in the revised Supplementary Fig 1. as well.

2. Explain how the cell counts were performed using a flow cytometer- did the authors use counting beads?

Response. Total number of cells isolated from the lungs following enzymatic digestion were counted using Cellometer 2000 (Nexcelom), and then percentages of specific gated populations from flow cytometry data were used to calculate the actual cell counts.

3. FACS plots are in low resolution.

Response. We have revised the figures with higher quality flow plots.

4. On pg. 5, the authors claim:

‘Individually, both LPS and PA14 induced significant increases in the numbers of CD11b+Ly6G+ neutrophils and CD11b+CD64+CD24- interstitial macrophages (IMs), both cell types having high phagocytic functions, with neutrophil numbers being only slightly higher in the LPS+PA14 mice (Fig. 2a, b).’

The figures don't show an increase of CD11b+CD64+CD24- interstitial macrophages (IMs) in PA14.

Response. We apologize for the confusion. We have revised the sentence in the text to read “Compared to PA14 mice, the lungs of both LPS only and LPS+PA14 mice showed significantly higher numbers of CD11b+Ly6G+ neutrophils and CD11b+CD64+CD24- interstitial macrophages (IMs) were detected, both cell types having high phagocytic functions. Neutrophil numbers were only slightly higher in the LPS+PA14 mice compared to that in LPS only mice (Fig. 2a).”

5. On pg. 7 the authors write:

‘As depicted in the volcano plots, some genes were upregulated in multiple clusters which included...’ Should be violine plots and not volcano plots.

Response. Please note that the plots are indeed volcano and not violin plots, since here we discussed data in Supp. Fig. 3 and not Main Fig. 4b (violin plot).

6. It is difficult to see colorbar changes in figures (e. g. 3b, 4c) due to the scale of colors. Please make a more precise scales and make sure of color consistency (high and low) along the figures.

Response. Thank you for the suggestion. Color bars and scales have been revised to ensure consistency across all Figs.

7. Fig. 5e- is presumably from scRNAseq data; however, this is not mentioned in the test/fig. legend.

Response. It is mentioned on page 10 under Results and also in the legend to Fig. 5. The figure now appears as Fig 5f.

8. in supp table 3, 30-day mortality, n (%) for At-risk for ARDS appears as 10 (18.9), which is not in agreement with the number of dots in fig. 6

Response. We apologize for any confusion that the previous display of our Fig. 6 may have caused. The number of dots in Fig. 6 matches exactly the numbers shown in Supplemental Table 3 although we recognize that the small size of the individual dots in the boxplots may have caused the confusion. We now provide revised boxplots with increased size of dotplots (organized with the `geom_beeswarm` function of `ggplot2`) to allow easy visualization of the individual data points and full graphical disclosure of our results.

9. In the discussion, the authors wrote:

‘We do not know whether the major source of LPS-induced *Ifng* was the N3 population.’ (page 15). This is followed by a paragraph that is not clear to me. Why the authors can't check *IFNg* expression in their single-cell data to indicate the relevant population?

Response. This is because we can detect very little *Ifng* transcript in any cell cluster although we see robust *IFN-γ* protein level in neutrophils by ICS. As we have cited, *IFN-γ* protein expression in neutrophils was also reported in the context of infection by *Streptococcus pneumoniae* and *Staphylococcus aureus* (Yamada et al., *AJRCCM*, 2011). As is true for many cytokine genes, post-transcriptional regulation of *IFN-γ* production has been reported although the precise underlying mechanisms have yet to be established (an in-depth review of post-transcriptional regulation of *IFN* genes was published a few years ago- PMID: 24702117). Our experimental system offers an excellent opportunity to address the mechanism. This is briefly discussed in Results.

10. In the discussion, please comment on how long the LPS effect lasts (the authors show 3-days, how about 1 week? 1 month?)

Response. We have conducted some preliminary experiments. The effect appears to last for up to 3 days after LPS exposure which interestingly coincides with the duration of *Aoah* expression.

11. Fig. 8 legend (the model) is very poor, as well as its description in the main text. The authors need to elaborate on their figure. E.g, what does the question mark stand for In the alveolus populations? It would help to indicate only the important findings in this illustration.

Response. The question mark is removed. We apologize for neglecting to include a detailed legend which is now corrected.

Reviewer #4

We appreciate the positive comments of the reviewer and our response to the questions raised is as follows:

1. While the authors provide an extensive description of the ScRNASeq data, the claim that the protective effects are mediated by N3 population is not substantiated, the authors failed to examine the IM2 population which is also induced in the pre-conditioned mice and carries a robust cell killing and phagocytic signature.

Response. We completely agree with the reviewer that IM populations in the lung tissue, especially IM2 and IM3, are also endowed with cell killing signatures like N3. However, an important difference between N3 and the IMs is the genes that constitute the cell killing pathway in the 3 populations. As shown in Fig. 4e, while *Cxcl9*, *Cxcl10* and *C3*, which also have potent anti-bacterial functions (cited in the MS), are common in all three populations, *Lgals3* is a component of this pathway only in N3. That is not to say that *Lgals3* is not expressed in the IMs. GSEA evaluates enrichment of a pathway and not an individual gene. The fact that *Lgals3* is a component of this enriched pathway in LPS+PA14 mice compared to PA14 mice suggests a specific role played by galectin-3 in cell killing in the N3 population in LPS-pre-exposed mice which may not be the case in IMs. Neutrophils were 100x more abundant in the BAL (airspaces) of LPS+PA14 mice compared to PA14 mice (Fig. 5i). Correspondingly, BAL bacterial load was significantly different between the two groups (Fig. 5). As discussed in the manuscript, studies have shown the importance of bacterial clearance in airspaces in both mice and humans and in this compartment neutrophils but not IMs (which are tissue-bound) would be involved in bacterial clearance. In *new* data, we show that BAL cells (>98% being neutrophils) in galectin-3 KO harbor significantly greater number of bacteria compared to BAL cells from WT mice (Fig. 7h). We further show that adoptive transfer of BAL cells (largely neutrophils) from LPS-exposed WT mice to LPS-exposed galectin-3 KO mice lowered the lung bacterial burden by ~ 1 log compared to that in mice that did not receive any cells (Fig. 7i). Taken together, while we do not discount the importance of IMs in bacterial clearance, the mouse and human data highlight an important impact of LPS in neutrophil training that results in a population N3. The unique cell killing pathway gene signature in N3 compared to all other neutrophil populations and the IMs guided us to focus on the molecule galectin-3 to assess its role in host protection in both mice and humans.

2. While the importance of neutrophil-mediated killing and bacterial clearance is not

denied, there are no specific studies to link the Galectin-3 findings with neutrophils. Ideally, deletion of Galectin 3 in neutrophils or macrophages would be needed to better delineate how Galectin-3 deficiency impairs host defense in the LPS-PA14 mice.

Response. Neutrophil-mediated bacterial killing in which galectin-3 plays a role, and for which we provide additional new data included in Fig. 7, is the novelty of our work. However, unfortunately, at this time, we are not able to provide data using galectin-3 conditional KO mice. Our new data show that BAL neutrophils isolated from LPS+PA14 galectin-3 KO mice have significantly higher bacterial burden than those isolated from similarly treated WT mice. We also discussed above the results of our adoptive transfer experiment.

3. The title of the paper claims Galectin-3 mediates protection from lethal bacterial infection in mice. Unless I am missing this data, I do not see evidence of improved survival in the mouse models in the absence of Galectin-3.

Response. Our data show that galectin-3 deficiency impairs bacterial clearance measured in the whole lung tissue and also in BAL neutrophils (new data) of the LPS+PA14 mice. While the galectin-3 KO LPS+PA14 mice consistently initially appeared lethargic and withdrawn in contrast to the WT LPS+PA14 mice which were active with no signs of sickness, they eventually recovered 24 h after infection. Our data suggest that the initial rapid reduction is facilitated by galectin-3 in WT LPS-exposed mice and potentially further aided by other molecules associated with the cell killing gene set such as *Cxcl9/10* and *C3* expressed by the N3 neutrophils. *Cxcl9/10* and *C3* also comprise cell killing gene sets in the IMs and may also have a role in bacterial clearance with galectin-3 being important early after infection. Thus, taken together, in the galectin-3 KO mice, it takes longer for mice to recover but our data suggest that additional molecules known to promote bacterial clearance are part of the bacterial elimination program.

Unquestionably, it will be interesting to study the dynamic aspect of bacterial clearance involving neutrophil and IM populations in both bacterial clearance and survival in future studies. It is also important to clarify that the mouse model we used was one of lethal bacterial infection as representative of what is encountered in ventilated critically ill patients that can result in VAP. Given that bacterial pneumonia can precipitate ARDS in humans, we sought to determine whether high neutrophil numbers and galectin-3 expression in available ETAs from patients with acute respiratory failure (ARF) associated with a positive outcome. Our data indeed show an association between ETA galectin-3 levels and survival of ARF patients, the data driven mostly by ARDS patients. With knowledge gained from additional experiments performed with the galectin-3 KO mice, we have revised the title of the manuscript to read **“Neutrophils and Galectin-3 Defend Against Lethal Bacterial Infection in Mice and Acute Respiratory Failure in Humans.”**

Comments:

1. Innate immune training has been classically described in monocytes and tissue macrophages. Have the authors considered examining the IM2 population which is also robustly induced by the LPS pre-conditioning. The specific focus on neutrophils while ignoring the IM2 population limits the strength of this manuscript.

Response. Thank you for referring to innate immune training. As mentioned in our response to reviewer #3, the topic of innate immune training has been largely limited to monocytes, which distinguishes our present study. However, as discussed above, we most definitely feel that the IMs are also involved in host-protection but it is beyond the scope

of this study to address their specific role. When viewed in aggregate, the data presented show that LPS-mediated training involves a marked increase in neutrophil numbers in the airspaces that can be recovered by BAL and that galectin-3 plays an important role in the reduced bacterial burden in these neutrophils.

2. The authors provide correlative data in the patient cohort and show elevated cell associated Galectin-3 in the LPS+PA14 mice. Can the authors do additional studies to help determine 1. The cellular source of the Galectin 3, 2. If the cell-associated Galectin-3 is secreted and adherent to the membrane, v intracellular v in the BAL (non-cellular fraction) in the mice. This would be important as Galectin-3 is thought to function as a DAMP and can be derived from numerous cell types. Studies to better delineate the specifics of Galectin-3 localization in the LPS-pre-conditioned model would help elucidate the mechanism of protection offered by Galectin-3.

Response. This is an important question since many cell types can indeed produce galectin-3. Galectin-3 has intracellular functions in bacterial killing in association with Gbp2 (which is expressed most robustly by N3 but also by IM2 and IM3 populations-Fig. 4) but is also secreted by cells which in turn can bind to the external surface of a cell (we have cited these studies). It is difficult to determine the relative contribution of the intra- vs extracellular protein. Having said that, we can detect BAL cell (largely neutrophils)-associated galectin-3 (Fig. 5n), which is significantly higher in the LPS+PA14 mice, these mice also having a much larger number (100x fold more) of neutrophils. Also, as mentioned above, BAL neutrophils from LPS+PA14 galectin-3 KO mice have considerably higher bacterial burden than those from similarly treated WT mice. Taken together, it is likely that intracellularly, the ability of galectin-3 to associate with Gbp2 to direct the latter to phagosome-containing bacteria (Feeley et PNAS, 2017) and also the direct bacteriostatic functions of galectin-3 (studies cited), contribute to efficient, rapid bacterial clearance. This function may play a central role in the survival advantage of ARF patients (further discussed below in light of additional analyses of human data-please see below).

3. Did exogenous Galectin-3 administration rescue the knockout mice?

Response. This is an important goal of ours but will need significant optimization and planned for the future.

4. Throughout the authors describe the BAL cell numbers and bacterial burden but with the exception of figure 1 showing differences in permeability and cytokine responses there are few other characterizations on the protective effect of LPS on PA14 infection. Can the authors provide some histology (if available).

Response. The primary protective effect is survival with 100% death without LPS and 100% survival when mice are pre-exposed to LPS. This has so far been reproduced in every experiment. Also, while as expected, LPS administration initially induces an inflammatory response to boost host defense (immune cell profiles using flow cytometry and scRNA-seq and lung histology (Supplementary Fig. 1a), the mice show complete resolution of inflammation at a later time of day 6 from initial LPS instillation = day 3 post-infection. As requested, representative lung histology is now included in Supplementary Fig. 1a.

5. Many of the studies were only done 2 times with few numbers of mice (3-4), Figure 1, additionally, Figure 7 demonstrating the effects of Galectin-3 deletion is only 1 study

with 4 mice. Please provide adequate numbers of mice and experimental replicates to ensure robustness of the data.

Response. All experiments have been repeated with reproducible results, as shown in the revised manuscript.

6. Using qFILM, the data demonstrates that LPS training increases phagocytosis by PMN. Would it be possible to examine the Galectin-3 KO mice using the same methodologies? Alternatively can the authors conduct ex-vivo studies demonstrating impaired phagocytosis of bacteria in the absence of Galectin-3 on neutrophils or macrophages.

Response. Based on its known functions, galectin-3 may be more important for bacterial killing than phagocytosis. As examples shown in our manuscript, the gene, *Fcgr1g*, which encodes the Fcγ receptor, and plays an important role in phagocytosis and innate immunity, is enriched in the N3 and IM2 clusters in the LPS+PA14 mice compared to that in the PA14 mice. Higher *Rac1* expression was also detected in N3 in the LPS+PA14 mice, *Rac1* being an important component of the plasma membrane Nox2 system in neutrophils. As mentioned in our response to reviewer #1, it is challenging to perform in vitro phagocytosis experiments with primary neutrophils that can truly reflect in vivo events.

7. There is an expansive literature on Galectin-3 and IFNγ regulation in tumor biology. How do the authors reconcile their findings of decreased IFNγ secreting PMN in the absence of Galectin-3 with the large body of evidence showing a role of Galectin-3 in dampening IFN? One would imagine there would be more IFNγ present in the absence of Galectin-3. While the authors show decreased intracellular IFNγ in the Cd11b/Ly6G/CD14 pmn it'd be interesting to see if systemic or lung levels of IFNγ are altered in the absence of Galectin-3.

Response. Galectin-3 can both augment and suppress IFN-γ production depending on context/disease state. As in our study, it can promote IFN-γ production in EAE and lupus models (PMID: 19124760; PMID: 29691398) but suppresses in cancer models as the reviewer has commented. Please see new Supplementary Fig. 8b showing expression of *Ifng* and other genes in WT and KO mice. The source of *Ifng* transcript detected in the whole lung is likely other innate cells such as NK cells.

8. The patient level data demonstrating the correlation between neutrophil elastase and galectin-3 may be misleading as this does not prove that the LRT Galectin 3 is derived from the neutrophils. Would try to rephrase the conclusions based on the correlative data. Would certainly remove the phrase “our data suggest a prognostic value of Galectin-3”. Prognostic of what, without additional data it'd be hard to claim that the ETA Galectin-3 is a prognostic biomarker in ARDS.

Response. We thank the reviewer for these helpful comments. We now provide more clinical data that demonstrate associations between LRT WBC markers and galectin-3 levels. We agree that we cannot assert a generalizable prognostic value to galectin-3 from available data, and we revised the corresponding statement in the Discussion section, per the suggestion of the reviewer:

“Although we found independent associations for neutrophil biomarkers in both clinical and research LRT biospecimens with galectin-3 levels, we could not prove that measured galectin-3 was indeed expressed and secreted by neutrophils. The associations between

LRT galectin-3 and survival in ARF patients are hypothesis-generating for a biologically and clinically relevant role of galectin-3 expression in the LRT of critically ill patients.”

9. There are several references made to post-viral COVID ARDS (in terms of hyperinflammation/IL-6 in the discussion). Would ask that the authors instead consider placing their findings in context of the large body of evidence examining the “hyperinflammatory” ARDS endotypes.

Response. Thank you for this very insightful comment, which led us to pursue additional analyses that advanced our understanding of galectin-3 in the context of host-response subphenotypes in ARDS.

We leveraged our existing data repository and classified patients in hyper- vs. hypo-inflammatory subphenotypes based on available biomarker values in our database.

We provide detailed methodological description of this classification and provide references to the model we used for classification.

In the Methods section, we added the following sentence:

“From available plasma biomarkers analyzed previously, we used a validated logistic regression model for classifying a patient into a hyper- vs. a hypo-inflammatory subphenotype based on plasma levels of IL-6, sTNFR1, and serum bicarbonate, as previously described.”

We found that the prognostically adverse hyperinflammatory subphenotype had lower LRT galectin-3 levels, which was consistent with our original findings with lower galectin-3 levels in non-survivors.

We have added the following text in the Results section:

“We then examined for relationship of LRT galectin-3 with 30-day survival and host-response subphenotypes. Among all patients followed for 30-day survival, survivors had significantly higher LRT galectin-3 levels compared to non-survivors (Fig. 6a), and this difference was driven mostly by patients with ARDS (Fig. 6b). Similarly, we found that patients classified to the prognostically favorable hypoinflammatory subphenotype had higher LRT galectin-3 compared to patients classified to the prognostically adverse hyperinflammatory subphenotype (Figure 6c) Stratified by 30-day survival, we found a highly significant correlation between LRT neutrophil elastase and galectin-3 in survivors only (Fig. 6d, $r=0.54$, $p<0.0001$), an effect that was also driven primarily by patients with ARDS, as well as patients classified to the hypoinflammatory subphenotype (Fig. 6e-f).”

We then contextualized our findings with the large body of evidence of ARDS subphenotypes with the following section in the Discussion:

Host-response subphenotypes based on plasma biomarkers have shown robust prognostic value in multiple cohorts of critically ill patients. Our data now provide further insights into the pathobiology of these blood-based subphenotyping by highlighting that the prognostically adverse hyperinflammatory subphenotype is associated with significantly lower galectin-3 levels in the LRT. These results underscore the importance of a regulated inflammatory response contributing to host’s resilience against lethal infection.”

We hope that we have been able to satisfactorily address all the concerns. Revised text is shown in yellow highlight.

REVIEWER COMMENTS

Reviewer #1 (Remarks to the Author):

The authors have made a forthright effort to address the criticisms raised in the previous review. The revised manuscript has been significantly improved by the additional data. I am happy to recommend this for Nature Communications without further revision.

Reviewer #2 (Remarks to the Author):

I thank the authors for their consideration of the comments previously given. I appreciate the effort which has gone into addressing some of these comments, particularly the increase in experimental replicates. I do think the story is very interesting and has exciting potential. However, there are still aspects of the work which need addressing.

Comment #1: I still have some issue with the lack of an LPS only control, particularly in the scRNA-seq experiment. The two groups chosen for the sc-RNA seq represent 2 completely different stages of infection, neutrophil recruitment, and maturation. In one group the lung has been 'inflamed' for 3 days + 4 h and the other for only 4 h. In light of the terminology used throughout the paper, i.e.:

Line 164 - LPS-promotes specific neutrophil and IM populations as revealed by scRNA-seq

Not showing the effect of LPS alone weakens the conclusions. It is clear from figure 1 that there are differences in the between the 'protection' of LPS exposure for 1 day vs 3 days – why? Is it related to neutrophil maturation? To claim the LPS is protective would require some analysis of this.

Comment #2 – The Aoah induction is indeed interesting, but why is this more pronounced at day 3 compared to day 1 LPS exposure – have the neutrophils had more time to 'prime' for this response. Is day 3 the key timepoint? From the data shown, this again suggests a process of maturation.

Comment 3. I appreciate the effort that has gone into increasing the n numbers in specific experiments. Annotating these changes in the revised manuscript would have been helpful to understand where the new numbers had been added, and how they may have affected the results.

Minor comments:

There are places in the manuscript where the wording or terminology is not clear, it could benefit from careful proof reading.

e.g.

Figure legends:

Line 598– the authors refer to 'two groups of mice' yet there are 4 different experimental groups and 3 independent experiments?

Statements are made in the manuscript which are not really supported by the data:

Line 35: LPS induced training of the innate immune system led to expansion of a neutrophil and an interstitial macrophage population distinguishable from other immune cells by being enriched in gene sets for phagocytosis- and cell-killing-associated genes.

Line 69: LPS exposure induced accumulation of a neutrophil population uniquely enriched in both phagocytosis-associated genes and a cell-killing gene set comprising *Lgals3*, which encodes the antibacterial protein, galectin-3.

Line 274: Pre-treatment with LPS helps to educate neutrophils for phagocytosis and cell killing

All these comments suggest LPS itself is having the effect but in the vast majority of cases the data appears to be coming from the PA vs LPS PA comparison, not the LPS.

This is a very interesting line of investigation and the authors appear to have the tools to address it.

Reviewer #3 (Remarks to the Author):

The authors have answered our major comments. We suggest including the following in the final version of the manuscript:

1. Please include the figure that the authors added to the rebuttal letter in point 2 (pg. 7 in the rebuttal letter) to the supplementary figures of the revised manuscript. Also, please mention in the text that LPS_PA_14_3 is different than LPS_PA_14_1 and LPS_PA_14_2. This is acceptable but has to be addressed.
2. On pg. 8 in 'c' in the rebuttal letter: it is still difficult to identify the populations of fig. S2a, in fig. S2ab. On supp fig. 2a, there are 18 populations; indicated by numbers. In supp fig. 2b there are 14 populations, indicated by names. To resolve, please indicated in S2b next to the names, their respective numbers as they appear in sup fig. 2a.
3. New fig 6- please add in a supp figure the data that was used to classify the patients into a hyper- vs. hypo-inflammatory subsets. The authors explain the parameters that were used. However, the actual data and the differences between hyper- and hypo-inflammatory are not shown.
4. Please add the response to minor comment 2 regarding cell counts (pg. 9 in the rebuttal letter) to the methods
5. The legend of fig. 8:
 - a. The question mark is still in the figure
 - b. The legend now includes details that do not appear in the scheme. We suggest writing a more concise

and clear legend for this figure.

Reviewer #4 (Remarks to the Author):

The authors have satisfactorily addressed my concerns.

The addition of the adoptive transfer BAL studies (Figure 7) greatly strengthens the manuscript. The addition of ARDS endotypes and reframing the patient level findings is also a strength.

Point-by-point response to the reviewers' comments

Revised text in this version of the manuscript is shown in green highlight

Reviewer #1 (Remarks to the Author):

The authors have made a forthright effort to address the criticisms raised in the previous review. The revised manuscript has been significantly improved by the additional data. I am happy to recommend this for Nature Communications without further revision.

Response. Thank you for your positive feedback and recommendation.

Reviewer #2 (Remarks to the Author):

I thank the authors for their consideration of the comments previously given. I appreciate the effort which has gone into addressing some of these comments, particularly the increase in experimental replicates. I do think the story is very interesting and has exciting potential. However, there are still aspects of the work which need addressing.

Response. We thank the reviewer for the enthusiasm expressed for our findings.

Comment #1: I still have some issue with the lack of an LPS only control, particularly in the scRNA-seq experiment. The two groups chosen for the sc-RNA seq represent 2 completely different stages of infection, neutrophil recruitment, and maturation. In one group the lung has been 'inflamed' for 3 days + 4 h and the other for only 4 h. In light of the terminology used throughout the paper, i.e.:

Line 164 - LPS-promotes specific neutrophil and IM populations as revealed by scRNA-seq
Not showing the effect of LPS alone weakens the conclusions. It is clear from figure 1 that there are differences in the between the 'protection' of LPS exposure for 1 day vs 3 days – why? Is it related to neutrophil maturation? To claim the LPS is protective would require some analysis of this.

Response. We understand the reviewer's comment that the 2 groups of mice subjected to scRNA-seq analysis experienced different exposures. We planned our scRNA-seq experiment based on the mouse survival and bacterial clearance data (Fig. 1) and the flow cytometry data (Fig 2a). First, we observed that a 3 day LPS exposure prior to PA14 infection led to a greater reduction in lung bacterial load than a 1 day exposure (Fig. 1b). We next found that LPS exposure alone over a 3 day period led to a significant increase in the numbers of neutrophils and IMs in the lungs of the mice (Fig. 2a). Collectively, the data revealed that the protective effect induced by LPS-priming was not instantaneous. Therefore, the data informed us that LPS induced expansion of neutrophils and IMs over a 3 day period and analysis of the lung cells at single cell resolution would allow a better understanding of these populations that expanded after LPS exposure. Since PA14 infection alone resulted in death of the mice while the 3 day LPS pre-exposure provided complete protection, our interest was to compare the 2 groups that had the same time frame of infection (4 h) differing only by pre-exposure to either PBS (PA14 group) or LPS (LPS+PA14 group) over a 3 day period. As already stated in the Results section, by 4 h after infection, mice not pre-exposed to LPS appeared visibly ill reaching criteria for euthanasia by 18-20 h after infection.

Given the reviewer's reservation with the subtitle of this section (line 164 in the previous version), we have revised it to state **"Pre-exposure to LPS in PA-infected mice promotes specific neutrophil and IM populations as revealed by scRNA-seq."** Please also see below similar revisions in sentences to convey this same message.

Comment #2 – The Aoah induction is indeed interesting, but why is this more pronounced at day 3 compared to day 1 LPS exposure – have the neutrophils had more time to ‘prime’ for this response. Is day 3 the key timepoint? From the data shown, this again suggests a process of maturation.

Response. Yes, that is precisely what our data suggest and also discussed in the manuscript. Aoah expression did not increase just by PA14 infection alone (after 4h) but its expression was highest when mice had been exposed to LPS for 3 days as compared to 1 day and the mice then infected for the same time period (4 h) by PA14 (Fig. 1f). This interesting profile of Aoah expression also aligned with the protection afforded by 3 day LPS pre-exposure and the increase in neutrophil numbers. Further detailed analysis of the role of Aoah in the protective function of LPS is planned for the next phase of this study.

Comment #3. I appreciate the effort that has gone into increasing the n numbers in specific experiments. Annotating these changes in the revised manuscript would have been helpful to understand where the new numbers had been added, and how they may have affected the results.

Response. In the previous revised version, we did update the figure legends for all presented data with new “n” numbers. Increasing the numbers strengthened our data and conclusions.

Minor comments:

There are places in the manuscript where the wording or terminology is not clear, it could benefit from careful proof reading.

e.g.

Figure legends:

Line 598– the authors refer to ‘two groups of mice’ yet there are 4 different experimental groups and 3 independent experiments?

Response. Thank you for alerting us about this oversight. The error has been corrected.

Statements are made in the manuscript which are not really supported by the data:

Line 35: In PA14-infected mice, LPS training of the innate immune system led to expansion of a neutrophil and an interstitial macrophage population distinguishable from other immune cells by being enriched in gene sets for phagocytosis- and cell-killing-associated genes.

Response. The sentence is revised to state “**Pre-exposure to LPS in PA-infected mice** trained the innate immune system as evidenced by increases in neutrophil and interstitial macrophage populations distinguishable from other immune cells by being enriched in gene sets for phagocytosis- and cell-killing-associated genes.”

Line 69: LPS exposure induced accumulation of a neutrophil population uniquely enriched in both phagocytosis-associated genes and a cell-killing gene set comprising Lgals3, which encodes the antibacterial protein, galectin-3.

Response. The revised statement reads “**Pre-exposure to LPS in PA infected mice** caused accumulation of a neutrophil population uniquely enriched in both phagocytosis-associated genes and a cell-killing gene set comprising Lgals3, which encodes the antibacterial protein, galectin-3.”

Line 274: Pre-treatment with LPS helps to educate neutrophils for phagocytosis and cell killing.

Response. The revised statement reads “**Pre-treatment with LPS in PA14 infected mice** helps to educate neutrophils for phagocytosis and cell killing.”

All these comments suggest LPS itself is having the effect but in the vast majority of cases the data appears to be coming from the PA vs LPS PA comparison, not the LPS.

This is a very interesting line of investigation and the authors appear to have the tools to address it.

Response. As stated in our response above, our goal was to address what distinguished mice that had been infected by PA14 for the same length of time but differed by LPS pre-exposure over a 3 day period that caused a drastic reduction in bacterial load and provided complete protection from death. We very much appreciate the reviewer's comments and we hope the results of the next phase of this study will provide additional insights into the protective function of LPS such as induction of molecules Aoah and AW112010.

Reviewer #3 (Remarks to the Author):

The authors have answered our major comments. We suggest including the following in the final version of the manuscript:

1. Please include the figure that the authors added to the rebuttal letter in point 2 (pg. 7 in the rebuttal letter) to the supplementary figures of the revised manuscript. Also, please mention in the text that LPS_PA_14_3 is different than LPS_PA_14_1 and LPS_PA_14_2. This is acceptable but has to be addressed.

Response. The Figure is included as a new Supplementary Fig. 3. Also, the the difference in the profile of mouse 3 in the LPS+PA14 groups as compared to the other two in this group is mentioned in the text.

2. On pg. 8 in 'c' in the rebuttal letter: it is still difficult to identify the populations of fig. S2a, in fig. S2ab. On supp fig. 2a, there are 18 populations; indicated by numbers. In supp fig. 2b there are 14 populations, indicated by names. To resolve, please indicated in S2b next to the names, their respective numbers as they appear in sup fig. 2a.

Response. Yes, we do recognize this issue and we have included a cluster annotation table as Supplementary Fig. 2b.

3. New fig 6- please add in a supp figure the data that was used to classify the patients into a hyper- vs. hypo-inflammatory subsets. The authors explain the parameters that were used. However, the actual data and the differences between hyper- and hypo-inflammatory are not shown.

Response. Thank you for this opportunity to clarify the phenotype classification approach and illustrate the clinical and biomarker group differences between these two patient subsets.

Per the reviewer's suggestion, we have added a Supplementary Fig. 7 that provides details on the statistical method and input data used to classify patients into hyper vs. hypoinflammatory phenotypes, as well as detailed comparisons of both biomarker and clinical variable data comparisons between the two phenotypes. We have also updated the text and methods sections accordingly.

4. Please add the response to minor comment 2 regarding cell counts (pg. 9 in the rebuttal letter) to the methods

Response. The following sentence has been added to the Methods section:

"Following enzymatic digestion of lung tissue, the total number of cells isolated from the lungs of each mouse was counted using Cellometer 2000 (Nexcelom), and then percentages of specific gated cell populations in flow cytometry data were used to calculate the actual cell counts."

5. The legend of fig. 8:

a. The question mark is still in the figure –

Response. We regret this error-the schematic has been edited.

b. The legend now includes details that do not appear in the scheme. We suggest writing a more concise and clear legend for this figure.

Response. The legend has been revised to more closely match the schematic.

Reviewer #4 (Remarks to the Author):

The authors have satisfactorily addressed my concerns.

The addition of the adoptive transfer BAL studies (Figure 7) greatly strengthens the manuscript. The addition of ARDS endotypes and reframing the patient level findings is also a strength.

Response. Thank you for your positive feedback.